# Extracting Local Reasoning Chains of Deep Neural Networks

**Haiyan Zhao**                                                    *Haiyan.Zhao-2@student.uts.edu.au*
*University of Technology Sydney*

**Tianyi Zhou**                                                            *zhou@umiacs.umd.edu*
*University of Maryland*

**Guodong Long**                                                        *guodong.long@uts.edu.au*
*University of Technology Sydney*

**Jing Jiang**                                                              *jing.jiang@uts.edu.au*
*University of Technology Sydney*

**Chengqi Zhang**                                                      *Chengqi.Zhang@uts.edu.au*
*University of Technology Sydney*

**Reviewed on OpenReview:** *https://openreview.net/forum?id=RP6G787uD8*

## Abstract

We study how to explain the main steps of inference that a pre-trained deep neural net (DNN) relies on to produce predictions for a (sub)task and its data. This problem is related to network pruning and interpretable machine learning with the following highlighted differences: (1) fine-tuning of any neurons/filters is forbidden; (2) we target a very high pruning rate, e.g., $\geq 95\%$, for better interpretability; (3) the interpretation is for the whole inference process on a few data of a task rather than for individual neurons/filters or a single sample. In this paper, we introduce NeuroChains to extract the local inference chains by optimizing differentiable sparse scores for the filters and layers, which reflects their importance in preserving the outputs on a few data drawn from a given (sub)task. Thereby, NeuroChains can extract an extremely small sub-network composed of critical filters exactly copied from the original pre-trained DNN by removing the filters/layers with small scores. For samples from the same class, we can then visualize the inference pathway in the pre-trained DNN by applying existing interpretation techniques to the retained filters and layers. It reveals how the inference process stitches and integrates the information layer by layer and filter by filter. We provide detailed and insightful case studies together with several quantitative analyses over thousands of trials to demonstrate the quality, sparsity, fidelity and accuracy of the interpretation. In extensive empirical studies on VGG, ResNet, and ViT, NeuroChains significantly enriches the interpretation and makes the inner mechanism of DNNs more transparent.

## 1 Introduction

Deep neural networks (DNNs) greatly reshape a variety of tasks — object classification, semantic segmentation, natural language processing, speech recognition, robotics, etc. Despite theirs success on a vast majority of clean data, DNNs are also well-known to be sensitive to small amounts of adversarial noises. The lack of sufficient interpretability about their success or failure is one major bottleneck of applying DNNs to important areas such as medical diagnosis, public health, transportation systems, financial analysis, etc.

Interpretable machine learning has attracted growing interest in a variety of areas. The forms of interpretation vary across different methods. For example, attribution methods (Bach et al., 2015; Sundararajan et al., 2017; Shrikumar et al., 2017; Montavon et al., 2017; Kindermans et al., 2017; Smilkov et al., 2017)

produce the importance score of each input feature to the output prediction for a given sample, while some other methods (Zeiler & Fergus, 2014; Simonyan et al., 2013; Erhan et al., 2009) aim to explain the general functionality of each neuron/filter or an individual layer regardless of the input sample. Another line of works (Ribeiro et al., 2016; Wu et al., 2018; Hou & Zhou, 2018) explain DNNs in a local region of data space by training a shallow (e.g., linear) and easily interpretable model to approximate the original pre-trained DNN on some locally similar samples. Thereby, they reduce the problem to explaining the shallow model. These methods essentially reveal the neuron to neuron correlations (e.g., input to output, intermediate layer/neuron to output, etc), but they cannot provide an overview of the whole inference process occurring inside the complicated structure of DNNs.

In this paper, we study a more challenging problem: *Can we unveil the major hidden steps of inference in DNNs and present them in a succinct and human-readable form?* Solving this problem helps to answer many significant questions, e.g., which layer(s)/neuron(s) plays the most/least important role in the inference process? Can two similar samples share most inference steps? Do all samples need the same number of neurons to locate the key information leading to their correct predictions? Do DNNs rely on entirely different neurons or filters for different (sub)tasks? Some of them are related to other problems such as network pruning (Han et al., 2015; Li et al., 2016) and neural architecture search (NAS) (Zoph & Le, 2017). For example, are winning tickets (Frankle & Carbin, 2018; Liu et al., 2018) universal over different tasks or classes? Does the weight sharing scheme in recent NAS methods (Pham et al., 2018; Liu et al., 2019a; Ying et al., 2019) limit the searching space or quality?

We develop an efficient tool called NeuroChains to extract the underlying inference chains of a DNN only for a subtask. A subtask of a classification task can be defined as classification on a small subset of classes, which in this paper refers to a few data drawn from a subset of 1000 classes in ImageNet. NeuroChains aims to extract a much smaller sub-network composed of a subset of neurons/filters exactly copied from the original pre-trained DNN and whose output for data from the subtask is consistent with that of the original pre-trained DNN. While the selected filters explain the key information captured by the original pre-trained DNN when applied to data from the same task/class, the architecture of the sub-network stitches these information sequentially, i.e., step by step and layer by layer, and recover the major steps of inference that lead to the final outputs. We parameterize the sub-network as the original DNN with an additional score multiplied to each filter/layer's output featuremap. Thereby, we formulate the above problem of sub-network extraction as optimizing differentiable sparse scores of all the filters and layers to preserve the outputs on all given samples.

The above problem can be solved by an efficient back-propagation that only updates the scores with fixed filter parameters. The objective is built upon the Kullback–Leibler (KL) divergence between the sub-network's output distribution and that of the original pre-trained DNN, along with an $\ell_1$ regularization for sparse scores over filters. We further use a sigmoid gate per layer to choose whether to remove the entire layer or block. The gate plays an important role in reducing the sub-network size since many data and subtasks do not rely on all the layers. We extract chains from the sub-network and visualize their filters and featuremaps of the original network by existing methods (Zeiler & Fergus, 2014; Erhan et al., 2009).

NeuroChains is a novel technique specifically designed for interpreting the local inference chains of DNNs. As aforementioned, it potentially provides an efficient tool to study other problems in related tasks. However, it has several **fundamental differences to network pruning and existing interpretation tasks**, which deter methods developed for these two problems from addressing our problem. **Comparing to network pruning**: (1) fine-tuning is not allowed in NeuroChains; (2) it targets a much larger pruning rate for succinct visualization, e.g., $\geq 95\%$ for VGG-19 (Simonyan & Zisserman, 2014), $\geq 99\%$ for ResNet-50 (He et al., 2016) and $\geq 92\%$ for ViT (Dosovitskiy et al., 2021; Touvron et al., 2021) on ImageNet with $\leq 200$ filters or patches remained; (3) it is for a few samples drawn from a task instead of the whole data distribution. **Comparing to mainstream interpretation tasks**: (1) NeuroChains produces an interpretation of the entire inference process for a specified task (e.g., a subset of classes) rather than of one neuron/filter or output on/around a single sample; (2) NeuroChains provides complementary information to the importance of individual neurons/filters for a task.

## 2   Related Works

**Interpretable machine learning** methods can be mainly categorized into the ones aiming to evaluate the importance of each input feature of a single sample and the ones explaining individual neurons/filters. Approaches in the first category usually rely on certain back-propagation from the DNN's output to derive an importance score for each input feature or hidden node. Earlier works are based on the back-propagated gradients, e.g., deconvolution (Zeiler & Fergus, 2014), back-propagation (Simonyan et al., 2013) and guided back-propagation (Springenberg et al., 2014). Sundararajan et al. (2017) proposed to (approximately) calculate the integral of the gradients along a path between a baseline point and the input sample, which ensures the sensitivity and implementation invariance lacking in some previous methods. More recent methods propose novel back-propagation rules to directly derive the attribution scores of neurons from output to input, e.g., DeepLIFT (Shrikumar et al., 2017), deep Taylor decomposition (Montavon et al., 2017), and layer-wise relevance propagation (LRP) (Bach et al., 2015).

Methods in the second class treat DNNs as black boxes and seek simple models to explain how the DNN's output changes in a local region. For example, Ancona et al. (2017) add perturbations to different parts of the input to evaluate how the perturbations change the output, which reflect the importance of different parts. Zeiler & Fergus (2014) covered different parts of the input with a gray square, which led to different prediction probabilities on the true class. Instead, Zintgraf et al. (2017) replaced each patch in the input image with the surrounding patch and tracked the induced changes in the output. In LIME, Ribeiro et al. (2016) trained a sparse linear model on noisy input-output pairs as a local surrogate approximating the original pre-trained DNNs, where the sparse weights are used to explain the importance of input features. As mentioned before, our main difference to the above methods are we explain DNNs for a (sub)task and its data and we further explain how DNNs step by step integrate the information of important filters/neurons.

**Network pruning** Han et al. (2015) and Li et al. (2016) remove redundant neurons/nodes or connections/weights from a pre-trained DNN and fine-tune the sub-network. Structural pruning removes whole layers/channels/filters/neurons according to a certain norm of the associated weights (Li et al., 2016) or sparsity (Hu et al., 2016). In contrast, Frankle & Carbin (2018) and Liu et al. (2018) prune a DNN during its training. Luo et al. (2017) apply pruning to two adjacent convolution layers at each time to take the dependency between the two layers into account. Liu et al. (2019b) and Guo et al. (2020) train sub-networks of different sizes in a large DNN at the same time to satisfy various constraints. Several recent works (Su et al., 2020; Evci et al., 2020) empirically verify "lottery ticket hypothesis", i.e., there exists sub-networks (i.e., winning tickets) that can reach comparable generalization performance as the original pre-trained DNN if re-trained. In contrast, the sub-network extracted by NeuroChains cannot be fully re-trained since it has to preserve the original pre-trained DNN's filters, and our goal is to retain the generalization performance only for a task with few data.

## 3   NeuroChains

### 3.1   Problem: Extract Sub-networks

Although the DNNs widely used nowadays are usually composed of hundreds of layers and millions to billions of hidden nodes. *When applied to samples from a subtask (e.g., composed of two classes), it is plausible that its inference process mainly relies on a small subset of layers and filters.* In this paper, we verify this conjecture by developing an efficient and practical algorithm, i.e., NeuroChains, to extract the subset and its underlying architecture as a sub-network whose filters are selected and exactly copied from the original pre-trained DNN while its outputs for a given subset of data or classes retain the ones produced by the original pre-trained DNN. Although DNNs are usually non-smooth in definition if using a non-smooth piecewise activation such as ReLU, when trained with the commonly used techniques, e.g., data augmentation, mix-up, dropout, the resulted DNNs are relatively smooth in a sufficiently small local region.

In order to preserve the original inference chains, we do not allow any fine-tuning or re-training on any filter or the weight vector corresponding to any neuron: they can only be exactly copied from the original pre-trained DNN. Let $F(\cdot; \{W^\ell\}_{\ell=1:L})$ (a mapping from input to output) denote the original pre-trained DNN, $W^\ell$ represents the set of filters/weight vectors in layer-$\ell$, and $W^\ell[i]$ represents the i$^{th}$ filter/weight

vector in layer-$\ell$. Any sub-network fulfilling our above requirement can be defined and parameterized by an indicator vector $M^\ell$ per layer, whose each entry is a $\{0, 1\}$ value indicating whether retaining the associated filter/neuron in $W^\ell$. We further define operator $\circ$ as

$$(W^\ell \circ M^\ell)[i] \triangleq \left\{ \begin{array}{ll} W^\ell[i], & M^\ell[i] = 1; \\ \vec{0}, & M^\ell[i] = 0. \end{array} \right. \tag{1}$$

Thereby, $\{M^\ell\}_{\ell=1:L}$ defines a qualified sub-network for inference chain and its weights are $\{W^\ell \circ M^\ell\}_{\ell=1:L}$, where we extend the operator $\circ$ to make $W \circ M = \{W^\ell \circ M^\ell\}_{\ell=1:L}$ given the original pre-trained DNN's weights $W = \{W^\ell\}_{\ell=1:L}$. Given a set of samples $\mathcal{X}$ drawn from a specific subtask, we can formulate the problem of finding an inference chain as the following combinatorial optimization, which aims to find the most sparse indicator $M$ (i.e., the sub-network with the fewest filters retained) that does not change the outputs of the original pre-trained DNN for $\forall x \in \mathcal{X}$, i.e.,

$$\min_{\{M^\ell\}_{\ell=1:L}} \sum_{\ell=1}^{L} \|M^\ell\|_1 \text{ s.t. } F(x; W) = F(x; W \circ M), \ \forall x \in \mathcal{X}. \tag{2}$$

However, it is impractical to directly solve this combinatorial optimization since the possible choices for $M^\ell$ is of exponential number. We relax the 0-1 indicator vector $M^\ell$ to a nonnegative-valued score vector $S^\ell$ of the same size. We define an operator $\odot$ applied to $W^\ell$ and its associated scores $S^\ell$ as

$$(W^\ell \odot S^\ell)[i] \triangleq S^\ell[i] \cdot W^\ell[i]. \tag{3}$$

Note we limit entries in $S^\ell$ within $[0, 1]$ due to the possible redundancy among filters in the original pre-trained DNN, i.e., there might be filters of similar functionality for the given samples and a preferred pruning should be able to only preserve one of them and multiply it by the number of those redundant filters in the sub-network. In addition, less constraints are easier to handle in optimization and helpful to find sub-network whose outputs are closer to that of the original pre-trained DNN, since the class of sub-networks with parameters $W \odot S$ includes all the sub-networks with parameters $W \circ M$. Hence, we relax the challenging combinatorial optimization to the following optimization, i.e.,

$$\min_{\{S^\ell\}_{\ell=1:L}} \frac{1}{|\mathcal{X}|} \sum_{x \in \mathcal{X}} l(F(x; W), F(x; W \odot S)) + \lambda \sum_{\ell=1}^{L} \|S^\ell\|_1, \tag{4}$$

where $l(\cdot, \cdot)$ is a loss function aiming to minimize the distance between the original pre-trained DNN's output $F(x; W)$ and the sub-network's output $F(x; W \odot S)$. In our experiments, for classification, we use KL-divergence between the output distributions over classes, where the two output distributions are computed by applying softmax to $F(x; W)$ and $F(x; W \odot S)$ respectively, i.e.,

$$l(F(x; W), F(x; W \odot S)) = D_{KL}(\text{softmax}(F(x; W)) \| \text{softmax}(F(x; W \odot S))). \tag{5}$$

In addition, empirical evidence (Krueger et al., 2017; Singh et al., 2016) show that for most samples there exist some layers that can be entirely removed without changing the final prediction. Hence, only a few hard and confusing samples need more delicate features, while most other samples can be correctly classified based on simple patterns from shallower layers. Therefore, in NeuroChains, we apply a sigmoid function with input score $\alpha^\ell$ as a gate $G^\ell$ determining whether to remove the entire layer-$\ell$ during pruning, i.e.,

$$G^\ell = 1 / \left[ 1 + \exp(-\alpha^\ell / T) \right], \tag{6}$$

where $T$ is a temperature parameter. With a gate $G^\ell$ applied after each layer-$\ell$ whose input and output has the same size (which is common in many DNNs), we can recursively define the input $H^{\ell+1}(\cdot)$ to the next layer-$(\ell+1)$, i.e., $H^{\ell+1}(x; \{W^{\ell'} \odot S^{\ell'}, \alpha^{\ell'}\}_{\ell'=1:\ell}) =$

$$\left\{ \begin{array}{ll} G^\ell \cdot F^\ell(H^\ell; W^\ell \odot S^\ell) + (1 - G^\ell) \cdot H^\ell(x; \\ \{W^{\ell'} \odot S^{\ell'}, \alpha^{\ell'}\}_{\ell'=1:\ell-1}) & \text{if input size = output size} \\ F^\ell(H^\ell; W^\ell \odot S^\ell) & \text{otherwise} \end{array} \right. \tag{7}$$

where $F^\ell(H^\ell; W^\ell \odot S^\ell)$ denotes the output of layer-$\ell$. The reason to use a gate here is that we expect to either remove the whole layer or retain it without adding an extra shortcut (which will change the original pre-trained DNN's architecture). Since we prefer to remove non-informative layers, in the objective, we add

another regularization $\alpha^\ell$ to encourage the removal of entire layers (because decreasing $\alpha$ reduces $G^\ell$ and thus increase the chance of layer removal). Therefore, the final optimization for NeuroChains is

$$\min_{\{S^\ell, \alpha^\ell\}_{\ell=1:L}} \frac{1}{|\mathcal{X}|} \sum_{x \in \mathcal{X}} l(F(x; W), H^{L+1}(x; \tag{8}$$

$$\{W^{\ell'} \odot S^{\ell'}, \alpha^{\ell'}\}_{\ell'=1:L})) + \lambda \sum_{\ell=1}^{L} \|S^\ell\|_1 + \lambda_g \sum_{\ell=1}^{L} \alpha^\ell,$$

Our objective above is similar to the one used in Network Slimming (Liu et al., 2017) but we optimize it for a subtask (so we can consider to remove layers) and we do not allow fine tuning on weights $W$.

## 3.2 Algorithm

Our algorithm is simply a standard back-propagation for the optimization problem in Eq. (8), which produces sparse scores for filters and gate values for layers. Note the weights in $W$ are fixed and the backpropagation only updates $S$. We initialize the filter scores $S = \vec{1}$ so $W \odot S = W$ at the beginning of optimization. We initialize the gate score $\alpha^\ell = 0$ for all $\ell = 1 : L$ so $G^\ell = 0.5$ at the beginning, i.e., the probabilities to remove or to retain a layer is equal. For classification, we set loss $l(\cdot, \cdot)$ to be the KL-divergence between the output distributions of the original pre-trained DNN and the sub-network. After convergence of the optimization, we then apply a simple thresholding to these scores to further remove more filters and layers: (1) we remove the filters with score under a threshold $\tau$; (2) we remove layer-$\ell$ if $G^\ell < 0.5$. This yields a sufficiently small sub-network architecture. Given a sub-network produced by NeuroChains, we then visualize its architecture and scores as the structure of the inference chains. Moreover, for samples from the same region, we visualize their inference pathway from the original pre-trained DNN by their activation patterns and featuremaps, respectively, using existing interpretation methods (Zeiler & Fergus, 2014; Erhan et al., 2009).

# 4 Experiments

| Statistics | ResNet-50 | VGG-19 |
|---|---|---|
| Top-1 test accuracy | 76.5% | 72.9% |
| Test images/sub-networks | 10000/1688 | 10000/1746 |
| Convolutional filters | 26560 | 4480 |
| Parameters of Conv-layers | 23454912 | 20018880 |
| Parameters of FC-layers | 2048000 | 123633664 |

| Statistics | ViT(DeiT-small) |
|---|---|
| Top-1 test accuracy | 79.9% |
| Test images/sub-networks | 10000/1500 |
| Patches across blocks | 2364 |
| Parameters of MHA-blocks | 7096320 |
| Parameters of FFN-blocks | 14178816 |

Table 1: Information of pre-trained DNNs in this paper.

In experiments, we apply NeuroChains to extract the inference chains of widely-adopted VGG-19, ResNet-50, and ViT which are all pre-trained on ImageNet. We provide the basic information of the three DNNs in Table 1. In the following, we will present **two quantitative analyses over hundreds of case studies**, which show that (1) NeuroChains is capable to produce sub-networks retaining only $< 5\%$ of filters and meanwhile preserve the outputs of the original pre-trained DNN in most cases; (2) the filters selected by NeuroChains with high scores are important to preserving the outputs since removing one will lead to considerable drop in performance. We also compare the capability of preserving the original neural network's outputs between NeuroChains and magnitude-based pruning and random pruning methods. We will then provide several detailed and insightful case studies and visualizations of extracted sub-networks for different subtasks.

## 4.1 Implementation Details

We implement NeuroChains by PyTorch (Paszke et al., 2017). In every case study, we firstly randomly sample 2 classes in ImageNet and then randomly sample 10 images from each class's images. Note that the sampled images may be wrongly classified to other classes by the original DNN. We apply inference on those 20 images and their outputs are used in solving the optimization of Eq (8) in order to extract the local

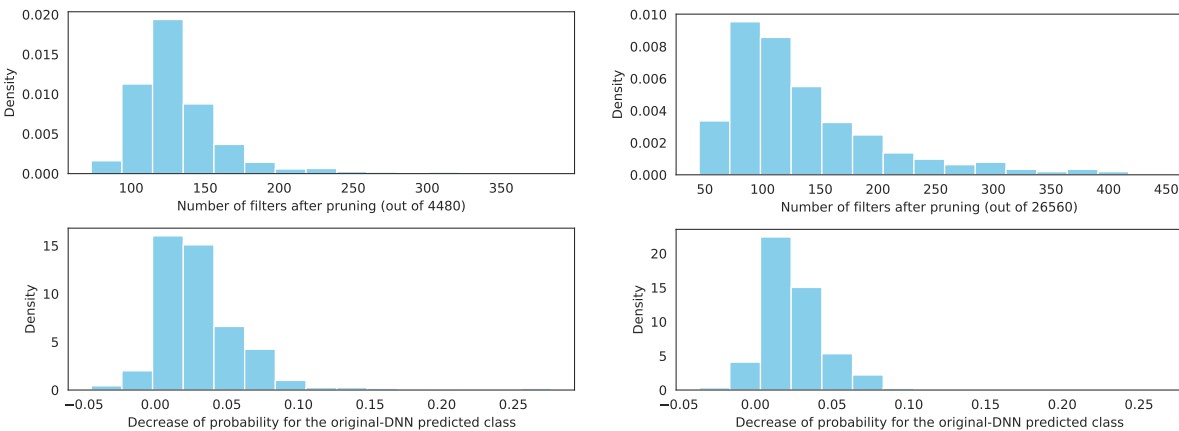

Figure 1: *Size and fidelity (how well the sub-networks preserve the original pre-trained DNN's outputs)* of 1500 sub-networks extracted by NeuroChains for VGG-19(left) and ResNet-50(right) in different case studies (tasks). The x-axis in the top plot refers to the number of retained filters, while the x-axis in the bottom plot is the decrease of probability on the original-DNN predicted class. It shows that NeuroChains can usually find very small sub-networks and meanwhile preserve the original pre-trained DNN's outputs.

inference chain in the form of a sub-network. For models with shortcuts, e.g., ResNet-50, the sigmoid gate is applied to prune a bottleneck block rather than a layer. A layer inside a block will be removed if the scores of all filters in the layer are nearly 0.

We use Adam optimizer for the optimization of Eq (8) for filter/layer scores. We use a fixed learning rate of 0.005. We set temperature $T = 0.2$ in sigmoid gate (Eq. (6)) to encourage the value $G^\ell$ close to either 0 or 1, and the threshold $\tau$ to goal scores is set to 0.1 so that the outputs of sub-networks are consistent. We only tried a limited number of choices on tens of experiments, and chose the best combination balancing the fidelity and sub-network size, and then applied it to all other experiments without further tuning. In particular, we tried $\tau \in \{0.01, 0.1, 0.5\}$, $\lambda \in \{0.001, 0.005, 0.01, 0.1\}$, and $\lambda_g \in \{1, 2, 5\}$. For different models, the weights of two penalties in Eq. (8) are different. For VGG-19, we use $\lambda = 0.005$ and $\lambda_g = 2$. While we choose $\lambda = 0.005$ and $\lambda_g = 1$ for ResNet-50. This choice performs consistently well and robust on all other experiments. The iteration steps of training is 300 and we stop training when the loss difference is quite small, i.e., less than 0.05. It costs only $\sim 90$s for VGG-19 and $\sim 55$s for ResNet-50 to extract a sub-network on a single RTX 6000 GPU since we only optimize a few number of scores.

### 4.2 Quantitative Analyses

Melis & Jaakkola (2018) propose some criteria to evaluate the interpretation methods for DNNs. In this paper, we extend some of their notations and present two quantitative analyses of NeuroChains over 1500 case studies for different subtasks, i.e., (1) **Fidelity**: does the sub-network preserve the original pre-trained DNN's outputs on the given samples? how does it change for sub-networks of different sizes? (2) **Faithfulness**: how well does the importance score of a filter reflect the degeneration on the fidelity caused by removing the filter from the sub-network? In this paper, we evaluate the fidelity and faithfulness by the decreasing amount of probability on the original pre-trained DNN's predicted class when using the sub-network for inference. All the above metrics are averaged over 1500 sub-networks and across all the images used to extract each sub-network.

Figure 1 shows the statistics of the fidelity for sub-networks of different sizes (measured by the number of filters) that are extracted by NeuroChains. Most sub-networks only retain $\leq 1\% (\leq 5\%)$ of ResNet-50(VGG-19) for succinct visualization but they preserve the outputs of ResNet-50(VGG-19) with high fidelity.

Figure 2 reports the faithfulness of extracted sub-networks, i.e., how a sub-network's performance in preserving the original pre-trained DNN's output degrade if removing one filter from it, and what is the relationship

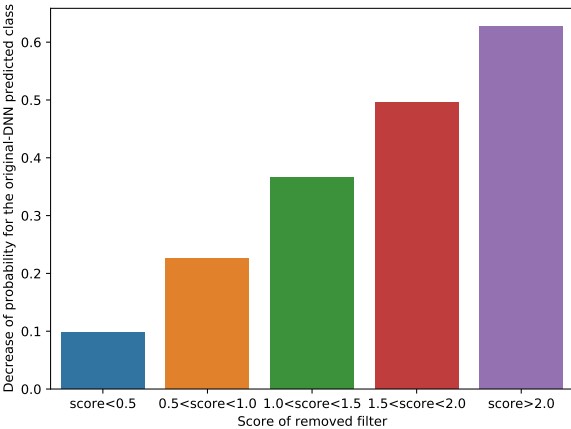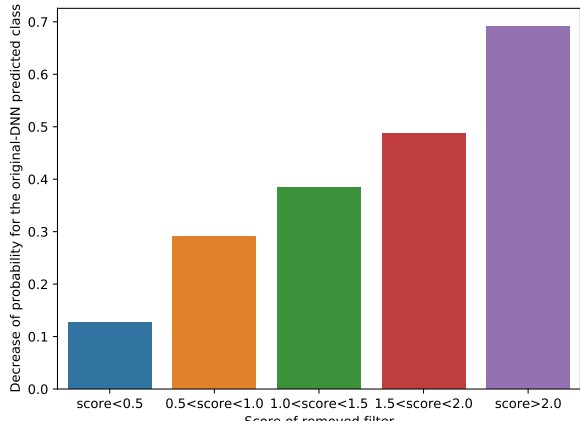

Figure 2: *Barplot of the faithfulness (filter score vs. fidelity degeneration caused by removing the filter)* of 783 sub-networks, each extracted by NeuroChains on 20 uniform samples randomly drawn from two classes for VGG-19(left) and ResNet-50(right). The x-axis refers to the interval of filter scores, while the y-axis denotes the decrease of the sub-network's probability on the original-DNN predicted class after removing a filter from the sub-network. It shows that the sub-networks suffer from more degeneration if removing a filter with higher score. Hence, the scores faithfully reflect the importance of filters in explaining the original pre-trained DNNs.

between this degeneration and the score of the removed filter. The statistics on 1500 sub-networks in Figure 2 show that as the scores of filters increase, the fidelity degrades more. The degeneration and the score are strongly and positively correlated, indicating that our optimized scores faithfully reflect the importance of filters in explaining the original pre-trained DNN. Moreover, removing even only one highly scored filter from the sub-network can significantly degrade the explanation performance. Hence, NeuroChains usually find the smallest sub-networks without redundancy among retained filters/layers, i.e., every critical inference step is retained.

We also present another faithfulness study based on a quotient metric defined below. Let $p, q \in \Delta^c$ ($\Delta^c$ is the probability simplex for $c$ classes) be the output probability vectors of the original neural net and the extracted sub-network respectively for same input. We define a quotient metric to measure the change of class prediction between $p$ and $q$, i.e.,

$$Q(p,q) = \frac{q[y] - \max_{z \in [c], z \neq y} q[z]}{p[y] - \max_{z \in [c], z \neq y} p[z]}, \quad y \in \arg\max_{z \in [c]} p[z], \tag{9}$$

where $y$ is the predicted class by the original neural net, and $Q(p,q)$ is the quotient of two probability differences computed respectively on the original neural net and the sub-network. In particular, it computes the difference of probabilities for class $y$ and the highest-rated other class. The sign of $Q(p,q)$ indicates whether the predicted class changes (e.g., it changes if $Q(p,q) < 0$) while the magnitude of $Q(p,q)$ measures the change in prediction confidence. The result is consistent with our above observations and is given in Figure 11 of Appendix.

We compare the capability of preserving the original neural network's outputs between NeuroChains and magnitude-based pruning (removing the filters whose output featuremaps' average magnitude (L2 norm) over all considered samples is small) and random pruning ("by chance") in Figure 3. In particular, under the same setting of each experiment in the paper, we prune the original VGG-19 and retain the filters with the largest featuremap magnitude in each layer or randomly, 180 in total (more than $157(mean) \pm 43(std)$ filters for sub-networks extracted by NeuroChains), and we then fine-tune the filters' scores. For Figure 3, the percentage of cases with KL-divergence$\leq 0.8$ for NueroChians, magnitude-based pruning and random pruning are **54.3%, 38.8%, 21.1%** respectively, and **9.3%, 19.4%, 38.8%** cases have KL-divergence$\geq 1.5$. Figure 3 shows the histogram of the KL divergence between the original output class distribution and the one produced by the sub-networks. For sub-networks generated by NeuroChains, the KL-divergence in most

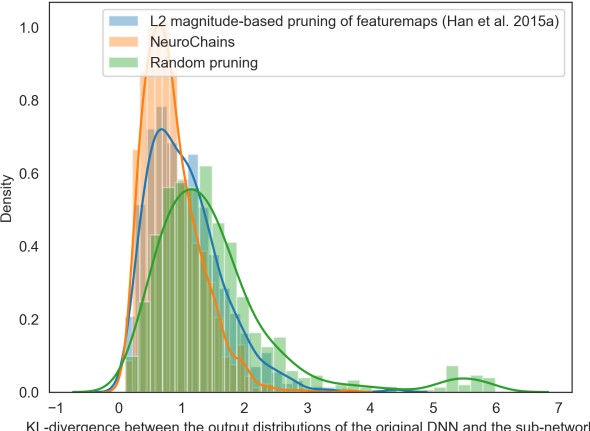

Figure 3: Comparison of different pruning methods on the capability of preserving the original network's output distribution (smaller KL divergence means better preservation) over $783{\times}20$ uniform samples.

cases stays close to 0, while the output preserving capability of simple pruning is much worse. More complete quantitative analyses for both VGG-19 and ResNet-50 can be found in the Appendix.

## 4.3 Case Studies

We present three case studies of the sub-networks extracted by NeuroChains. For Figure 4 and Figure 5, the data points are from classes which are easy to tell apart while in Figure 6 images are sometimes mis-classified. The case study of multi-classes classification task is shown in Figure 17 in Appendix. The visualization in each case study is composed of two parts: (1) the sub-network's architecture and filter scores; (2) original images from each class and the visualization of the image's NeuroChains extracted from the original pre-trained model. The true class and predicted class of the sample are shown above the image. They show that NeuroChains considerably enrich the explanation details of DNN's inference process. By connecting the important filters from different layers, the extracted sub-network highlights the main steps leading to the output prediction.

On the sub-network's architecture, we use "L0" to denote the corresponding convolution layer in VGG-19 and "L0_1" to denote the first filter from this layer. For ResNet-50, we further use "L1B1" to denote the first sub-block in the first bottleneck block, "SC" for the shortcut connection and "C1" for the first convolution layer in the sub-block. The redder the node in the sub-network, the larger the scaling score, conversely, the bluer the node, the lower the score.

NeuroChains are stitched by filters step by step and the visualization of each filter and the corresponding featuremap generated by the original pre-trained model are displayed in each step. The visualization of each selected filter is achieved by maximizing its activation w.r.t. the input. Afterward, we shows the patterns that the filter aims to detect which is independent of the input image. More case studies are given in Appendix.

## 4.4 Detailed Analysis of Case Studies

In Figure 4 and Figure 5, the case studies of VGG-19 and Resnet-50 are shown. For each sample from the subtask, two NeuroChains which present the inference process of the original pre-trained model are displayed.

In Figure 4, features extracted by the chains of strawberry gradually evolve from low-level to high-level, e.g., in the first chain, L1B1SC_177, L2B1SC_317, and L3B1C1_23 extract the low-level patterns of red-green color and texture, L3B1C2_18, L3B1C3_818, and L4B1C1_417 look like the abstract pattern of strawberry, and L4B1C2_328, L4B1C3_511 and L4B1SC_511 capture the pattern of the skin of strawberry. While the first chain of strawberry focus on the skin, the second chain extracts the pattern of the leaves,

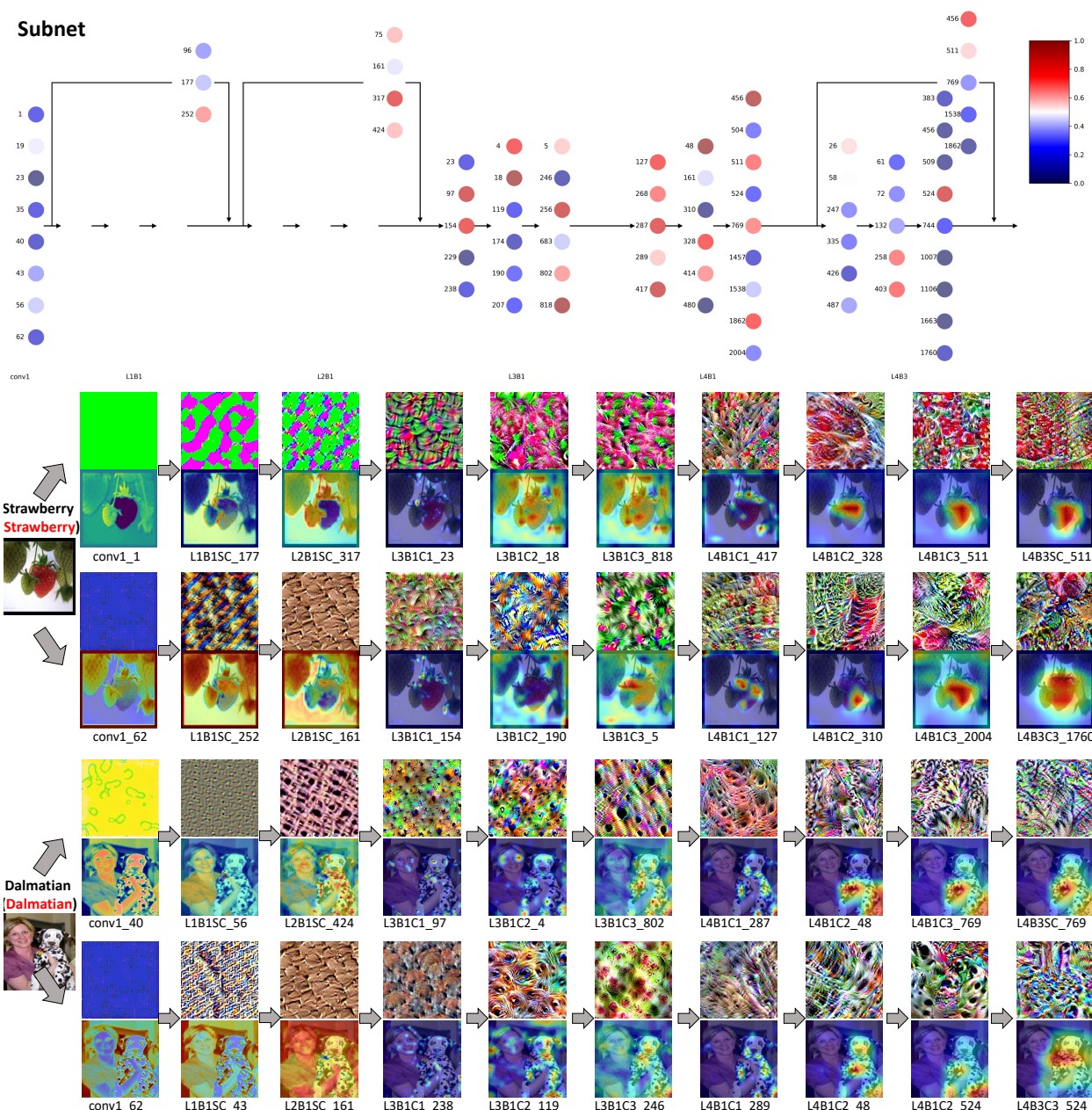

Figure 4: Inference chain by NeuroChains for ResNet-50 (pre-trained on ImageNet) when applied to 20 test images of "strawberry" and "dalmatian". The sub-network retains only 13/67 layers and 77/26560 filters of the ResNet-50. Refer to Section 4.4 for detailed analysis of extracted chains.

i.e., L3B1C1_154 and L3B1C3_5. The leaves in the featuremaps of L4B1C1_127 and L4B1C3_2004 are highlighted. L4B3C3_1760 extract the global patterns of strawberry. Both the two chains of dalmatian extract the global patterns of dalmatian's black and white fur. In shallow layers, L3B1C1_97, L3B1C1_238, L3B1C2_4, and L3B1C2_119 capture a more local black spot pattern of the dalmatian while L4B1C1_289, L4B1C2_48, L4B1C3_769 and L4B3C3_524 reveal the global patterns.

In Figure 5, the featuremaps show that the first chain of kangaroo extracts the pattern of eyes and noses, while the second chain pays more attention to the fur of kangaroos. L21_296, L23_393, L32_453, and L34_66 turn from a round pattern to an eye and nose pattern. L21_24, L32_463, and L34_188 look like

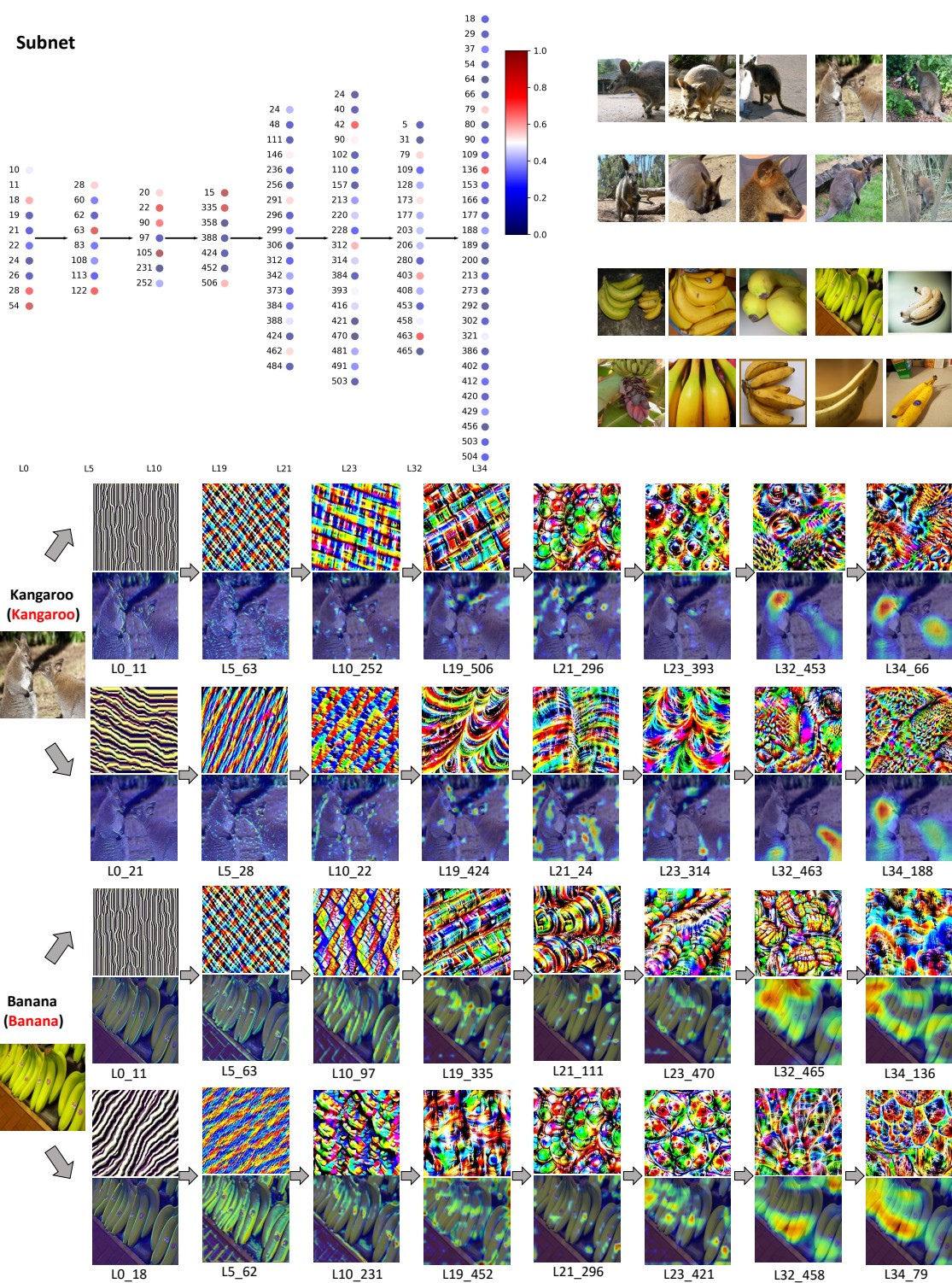

Figure 5: Inference chain by NeuroChains for VGG-19 when applied to images of "kangaroo" and "banana". The sub-network retains only 8/16 layers and 116/4480 filters of the VGG-19. The scores for selected filters are represented using the colormap on the top-right. Refer to Section 4.4 for detailed analysis of extracted chains.

the fur pattern of animals. In the first chain of bananas, L19_335, L21_111, L23_470, and L32_465 turn from the curved rectangle pattern to a hand of bananas. The L32_458 and L34_79 in the second chain of bananas seem like the pattern of toes and roots of bananas. The corresponding areas in the featuremaps are also highlighted. The two chains focus on bananas' global and local patterns, respectively.

In Figure 6, we first show the NeuroChains of correctly predicted castle and stone wall. Then, using the same filters from the extracted chains, we draw the featuremaps of a stone wall which is wrongly predicted as the castle by the original pre-trained model, to uncover the inner mechanism of false prediction. The featuremaps of the wrongly predicted stone wall from filters activated by castle show that the featuremaps in the mid-part of the chains are rarely activated, like L21_300 and L23_336, which is in line with our expectations. But in the deep layers, L32_15 and L32_45, which activate the roof and body of the castle, also highlight a wide area in the wrongly predicted stone wall image. This indicates that the wrongly predicted stone wall sample contains similar features as castle. The featuremaps of the wrongly predicted stone wall from filters activated by stone wall (correctly predicted) also shows the anomaly. L21_197, L25_26, and L32_260, which are critical to the correctly predicted stone wall, extract little information from the wrongly predicted stone wall. We find that L21_124 and L32_15 are both important to identify the castle and stone wall, which implies that castle and stone wall have similar patterns and are easily confused.

### 4.5 Applying NeuroChains to Vision Transformer

To show that NeuroChains can be applied to different types of neural networks for comparing their reasoning chains, we additionally apply NeuroChains to the vision transformer (ViT). Since recent work (Tang et al., 2022) discovered the effectiveness and advantages of patch pruning for ViT, our method produces a score for each patch in different blocks of ViT. In particular, NeuroChains freezes all the parameters in ViT and only optimizes the scores for each patch. At the end of patch pruning, patches with small scores are removed. The patch scores are not shared among training samples, i.e., each input sample has its own patch scores, so the critical patch for each sample can be preserved. In Fig. 7, we provide a case study of NeuroChains on ViT. The top figure show the patches (red squares) retained in each ViT block. In block1, the whole image is taken as the input. As the blocks get deeper, more patches are removed and only a few patches are retained in the last few blocks. The pruning results imply that the patch preserved in the last block contains the most critical information to classify the input sample.

To study the inference process of ViT on the given sample, we generate the attribution heatmaps for both the class token and patch token in different blocks. The attribution scores are computed as following:

For a ViT with $L$ blocks and $N$ patches, the attention score between patches in block-$\ell$ can be calculated by $\mathbf{Attention}_\ell = softmax(\frac{\mathbf{Q}_\ell \mathbf{K}_\ell^{\mathbf{T}}}{\sqrt{\mathbf{d}_{\mathbf{k}_\ell}}})\mathbf{V}_\ell$, where the $Q_\ell, K_\ell$ and $V_\ell$ are queries, keys and values in the multi-head attention module of block-$\ell$. The attribution of each patch in block-$\ell$ to the original patches in the input sample is computed as

$$\mathbf{Attribution}_\ell = \mathbf{Attention}_\ell \times \mathbf{Attribution}_{\ell-1} + \mathbf{Attribution}_{\ell-1}, \tag{10}$$

$\mathbf{Attribution_0} = I_N$, indicating that in the input all patches only attributes to themselves. Since $\mathbf{Attention}_\ell$ is calculated based on the output patches of the previous block, in Eq. 10, we multiply the $\mathbf{Attention}_\ell$ by $\mathbf{Attribution}_{\ell-1}$ to include the attribution to patches in the current block. We further add $\mathbf{Attribution}_{\ell-1}$ to reflect the shortcut operation in ViT blocks: the output patches of block-$(\ell-1)$, which encode the attribution of patches in block-$(\ell-1)$, are added directly to the output of block-$\ell$.

In the middle row of Fig. 7, we show the attribution heatmaps of the class token in different blocks of the pruned ViT. In the shallow blocks close to the input, it attends all patches with similar attribution scores. As the blocks gets deeper, the class token gradually focuses on the critical patches essential to classify the input image.

In this case study, only one patch token is retained in the last block of the pruned ViT. The attribution heatmaps of this patch are shown in the bottom row of Fig. 7. The patch token pays more attention to itself in the first few blocks but it starts to attend to more important patches in later blocks that are more relevant to the predicted class.

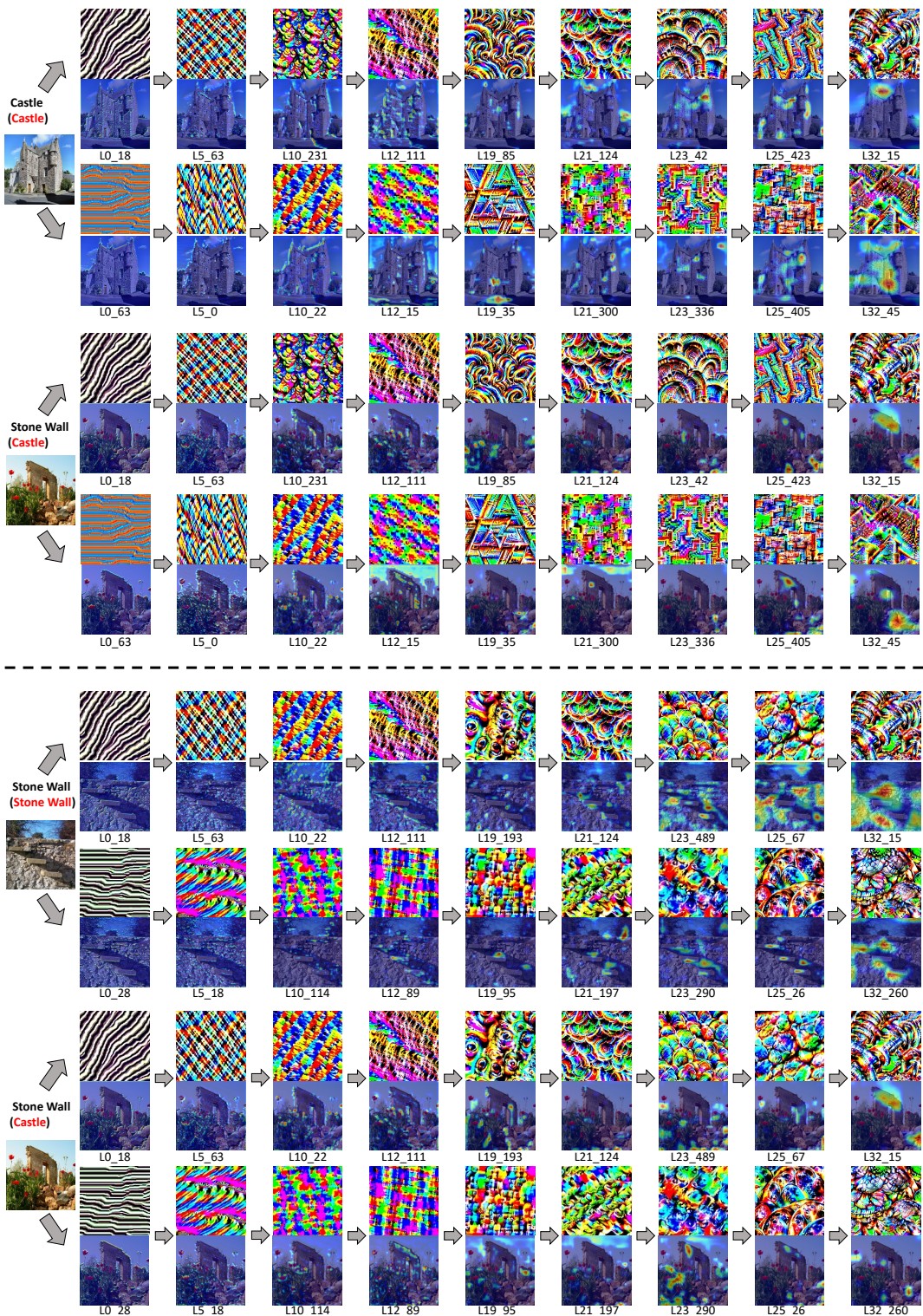

Figure 6: Inference chain by NeuroChains for VGG-19 when applied to images of "Castle" and "Stone Wall" which confuse DNN models and may be mis-classified. The sub-network retains only 10/16 layers and 121/4480 filters of the VGG-19. Due to space limitation, the subnet can be found in Appendix. Refer to Section 4.4 for detailed analysis of extracted chains.

The results of applying NeuroChains to ViT in Fig. 7 show that the inference process of ViT and CNN are very different. In CNN, from the shallow layers to the deep ones, the model first extracts the low-level local patterns of the input image and then aggregates them to get high-level global patterns step by step towards classifying the input image. In ViT, the model can capture both the local and global patterns of the input since the very bottom/early blocks/layers. During inference, the model gradually identifies patches important to the classification task and mainly extracts patterns relevant to the class token.

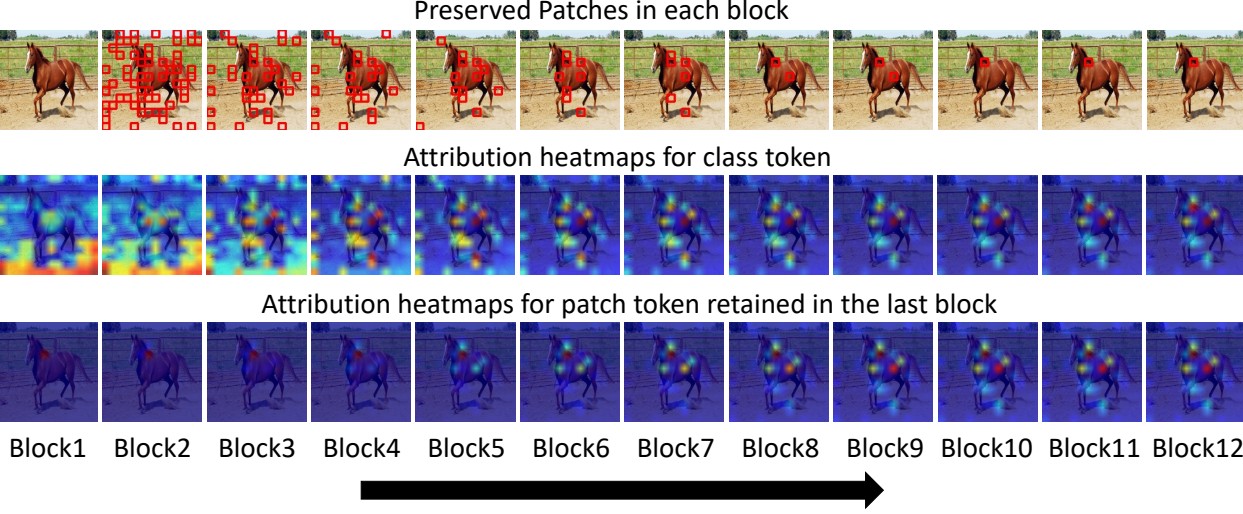

Figure 7: A case study of applying NeuroChains to vision transformer (ViT). Top: the retained patches (red squares) in each block. Middle: attribution of the class token to the original input patches. Bottom: attribution of the patch token (retained in the last block) to the original input patches.

## 5    Conclusion

We propose an efficient approach NeuroChains to extract the main inference chains/steps in a DNN when applied to a subtask. We learn a sparse scoring of filters/layers to extract a sub-network retaining a subset of filters/layers from the original pre-trained DNN. We provide case studies and quantitative analyses to demonstrate that NeuroChains produces more informative and reliable explanation to the inner inference process of DNNs. Since local reasoning chain extraction is a novel and challenging problem even for classification, we will investigate more complicated tasks like regression and detection in the future work.

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

# A  Appendix

## A.1  Ablation Studies of Layer Pruning

In Fig. 8, we compare the number of filters in sub-networks extracted with or without layer pruning. When layer pruning is applied, for both VGG-19 and ResNet-50, most sub-networks sizes lie in a small range 0 to 200. Without layer pruning, much more redundant filters will be preserved in the sub-networks. Hence, layer pruning is essential to build a small, human interpretable, and effective sub-networks.

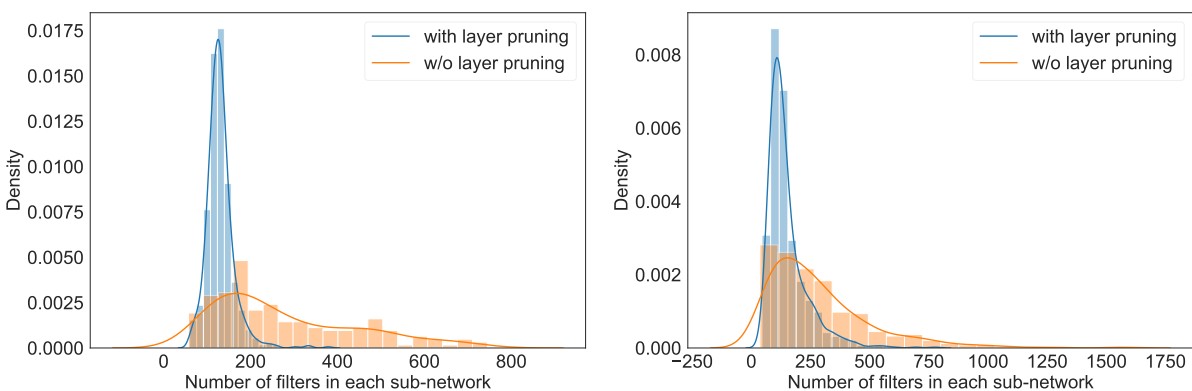

Figure 8: Comparison between number of filters in sub-networks extracted with or without layer pruning on VGG-19(Left) and ResNet-50(Right).

## A.2  Effect of the Number of Classes

In Fig. 17, we show a case study of applying NeuroChains to a 3-classes classification sub-task, demonstrating the capability and effectiveness of NeuroChains on multi-classes problem. However, as classes per sub-task increases, the number of preserved filters in the sub-networks also increases, which may weaken the interpretability. We test NeuroChains on the 10-classes classification sub-tasks. The averaged number of filters in the extracted sub-networks is 795(1206) for VGG-19(ResNet-50). Though much smaller than the original network, the extracted sub-networks are too large to explain.

That being said, NeuroChains can explain the correlation among multiple classes by identifying the chains important to distinguish a class from the others. We conduct a binary-classification experiment shown in Fig. 10. In this sub-task, samples in one class are all eagles, while samples in another class are randomly selected from different classes, as shown in the top right of the figure.

From the figure, the sub-network extracts two characteristic patterns for eagles. The first chain detects the pattern of feathers, while the second chain extracts the pattern of eyes and heads. These two patterns are

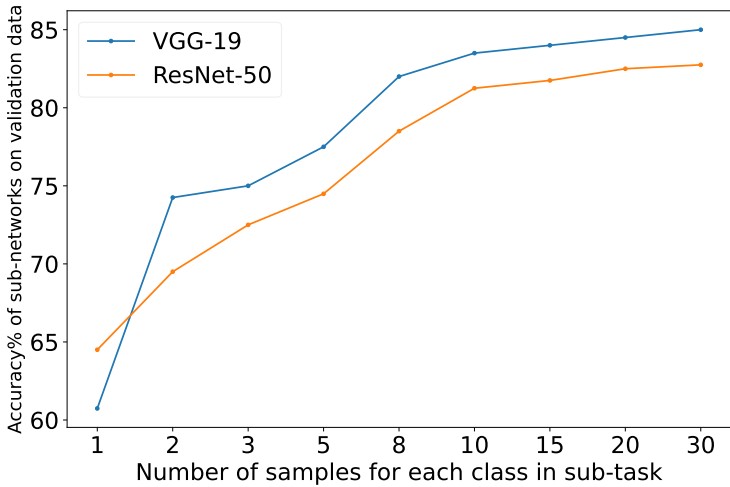

Figure 9: The validation-set accuracy of sub-networks vs. the number of training samples per class in each sub-task.

enough to recognize eagles. For images from other classes (a horse image in this case study), the key pattern of the object will be identified. In the figure, the first chain of the horse locates the elbow part. Filter $L30\_286$ contains both the patterns of eyes and elbows. However, in the second chain of the eagle, it only activates the eyes; in the chain of the horse, the elbow of the horse is highlighted. The unique patterns in other objects will be extracted in order to distinguish them from eagles. In the second chain of the horse image, we use the filters in the first chain of eagle to extract patterns for the horse image. The featuremaps show that the eagle filters are not activated on the horse image, so it can be distinguished from eagles.

### A.3 Effect of the Number of Training Data

Fig. 9 shows how the sub-networks' validation-set accuracy when increasing the training samples per class in each sub-task. For both pre-trained VGG-19 and ResNet-50, when the number of samples per class is too small ($\leq 5$), the sub-network tends to be overfitting and cannot generalize well to unseen validation data. However, when the number of samples per class increases to 10, the sub-network starts to achieve promising validation-set accuracy. If further increasing the samples per class, the validation-set accuracy quickly saturates. Therefore, in the experiments, 10 samples per class turn to be the sweet spot to extract sub-networks with sufficiently good generalization performance so they can be interpreted as the inference chains of the pre-trained model on given sub-tasks.

### A.4 Details of the Heatmaps of Features

To obtain the heatmaps, each filter applied in the sub-network produces a featuremap for an input image. By applying an activation function to the featuremap, only a few pixels are activated and they highlight the regions in the featuremap, which represent what features/patterns of the input sample are detected by the filter. Since the featuremap size is always smaller than the input image, we apply the bilinear interpolation to upsample the featuremap up to the input image size. Finally, we overlap the resized featuremap with the input image, which results in a heatmap highlighting the regions of the input image represented by the corresponding filter.

### A.5 More Quantitative Analysis

We performed 783 experiments each using 20 samples uniformly drawn from two classes. We evaluated these newly generated sub-networks using the quotient metric "A quotient of "diff to highest scoring other class (extracted)" / "diff to highest scoring other class (original)" Eq. (9). We visualized the result in Figure 11:

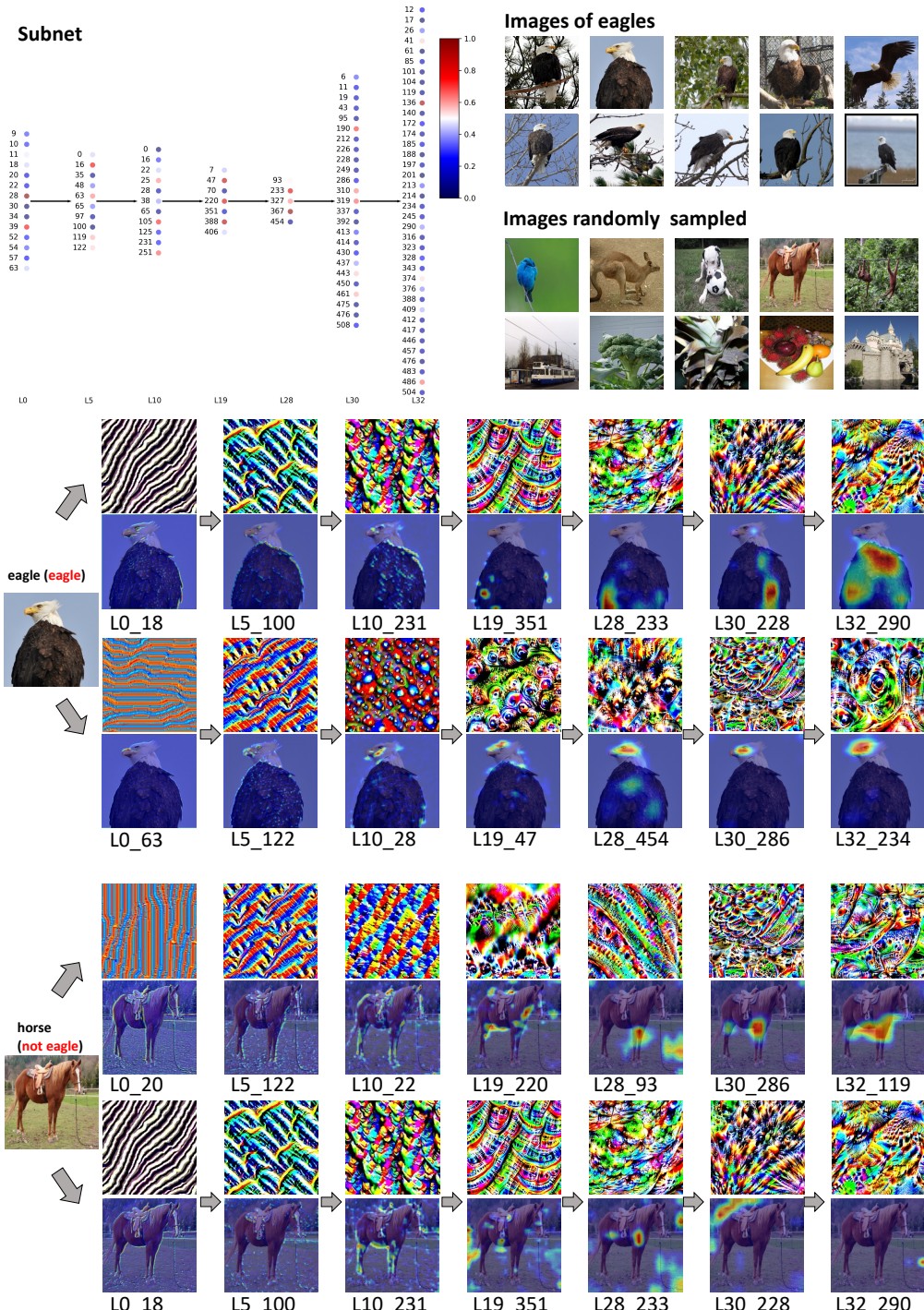

Figure 10: Inference chain by NeuroChains for VGG-19 when applied to tell apart images of "eagle" and other randomly selected samples. The sub-network retains only 7/16 layers and 110/4480 filters of the original VGG-19.

The left plot is the histogram of the quotient computed over all the 783×20 samples. The histogram shows that most samples keep the original predicted label after pruning, i.e., NeuroChains can preserve the original

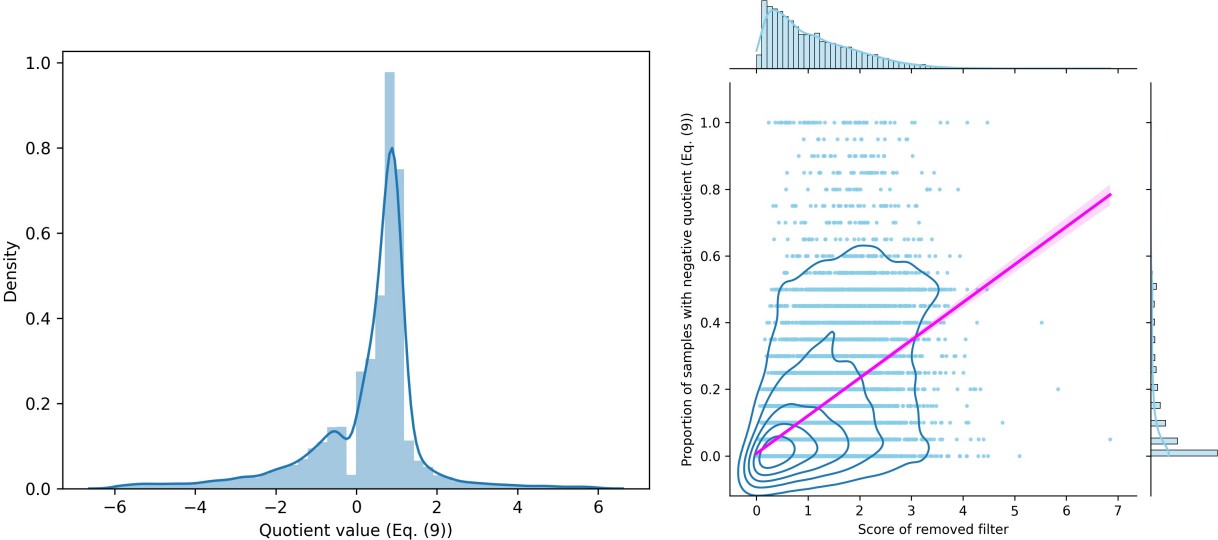

Figure 11: **Left:** Histogram of the quotient metric in Eq. (9) computed over all the $783\times20$ samples. **Right:** Faithfulness of NeuroChains in terms of the quotient's sign.

pre-trained DNN's outputs in most cases. Moreover, the number of filters preserved in these sub-networks is $157(mean) \pm 43(std)$, which is small enough to explain.

The right plot reports the Faithfulness of NeuroChains in terms of the quotient's sign. We remove each filter from each sub-network and report how many samples' predicted labels are changed after the removal, i.e., the quotient is negative. Each point in the scatter plot corresponds to a sub-network, the x-axis is the score of the removed filter given by NeuroChains, and the y-axis is the proportion of samples with negative quotients. The plot shows a strong linear correlation between the score of the removed filter and the degradation of faithfulness. Since removing filters with high scores results in more samples with predicted class changing after pruning, the score given by NeuroChains measures the importance of filters in DNN inference.

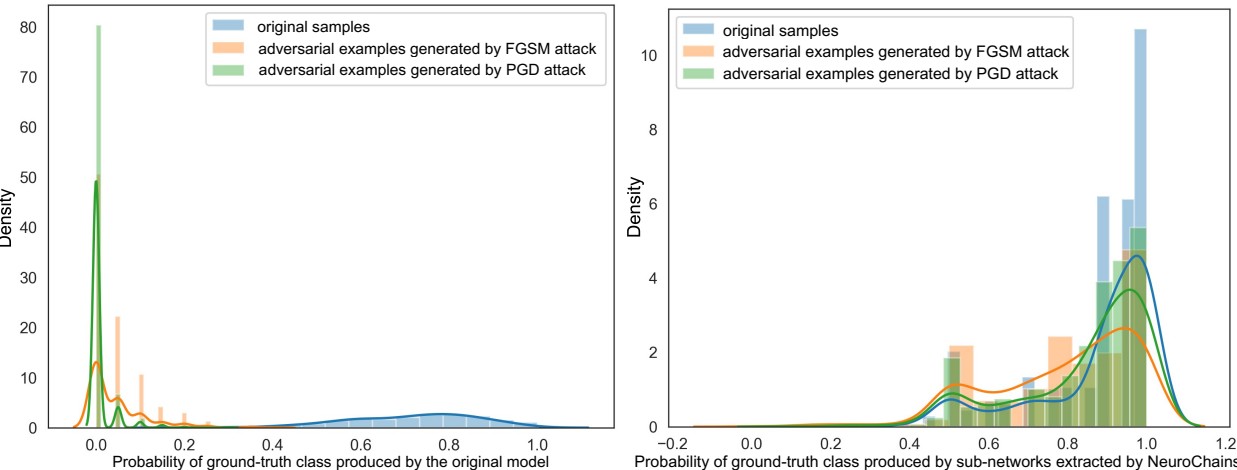

Figure 12: Robustness of original model (Left) vs. extracted sub-networks (Right) under different attacks.

In order to evaluate NeuroChains on the subtasks in the raw input space, we extract sub-networks for uniformly random drawn samples and then evaluate the sub-networks on these samples' adversarial examples generated by two types of attacks: fast gradient sign method (FGSM) (Goodfellow et al., 2014) and projected gradient descent (PGD)(Madry et al., 2018). Figure 12 compares the robustness of the original neural net

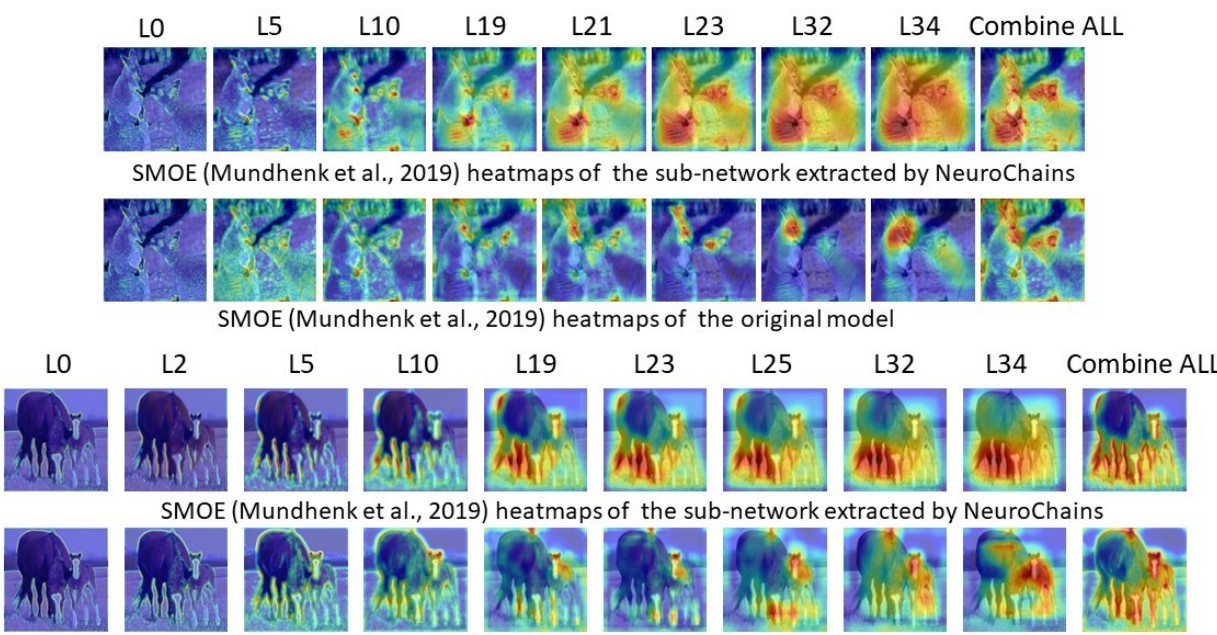

Figure 13: Case studies of SMOE generated heatmaps for the original network and the NeuroChains extracted sub-network.

(Left plot) and the extracted sub-networks (Right plot) under different attacks: each of the plots show the histogram of the output probability for the ground-truth class on those samples (original and adversarial). The left plot shows that the two types of adversarial attack are very effective on the original neural net in reducing the probability of ground truth class. In contrast, the right plot shows that the NeuroChains extracted sub-networks are much more robust to the attacks, because the optimization in NeuroChains not only removes the irrelevant filters but also strengthens the important filters by assigning them weights $> 1$. This demonstrates the effectiveness of NeuroChains when applied to subtasks with non-smooth raw-input space, and the extracted sub-networks in this case significantly improves the robustness of the original model in defending adversarial attacks.

In Figure 13, we show two case studies of comparing SMOE (Mundhenk et al., 2019) generated heatmaps for the original network and the NeuroChains extracted sub-network. We can see that the patterns extracted by the two networks are consistent and are all critical patterns for the class, e.g., the eyes and fists of kangaroos and the feet and face of the horse. However, compared with the original network, these patterns are strengthened in much shallower layers of the sub-network, producing better interpretations. This observation is also consistent with the result of analysis on adversarial attacks in Figure 12.

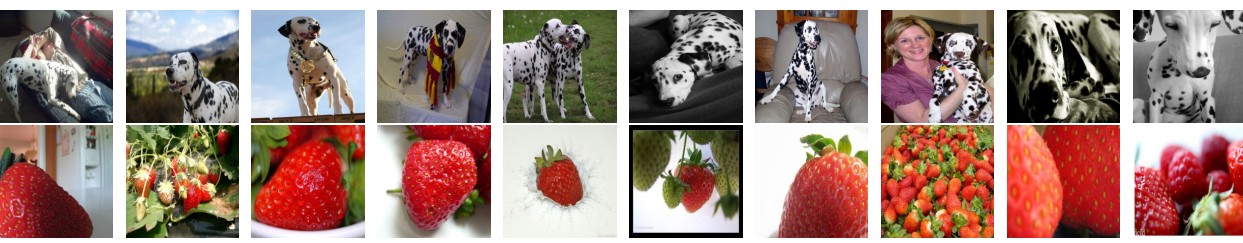

Figure 14: The 20 images from 2 classes used to extract the sub-network of case study in Figure 4.

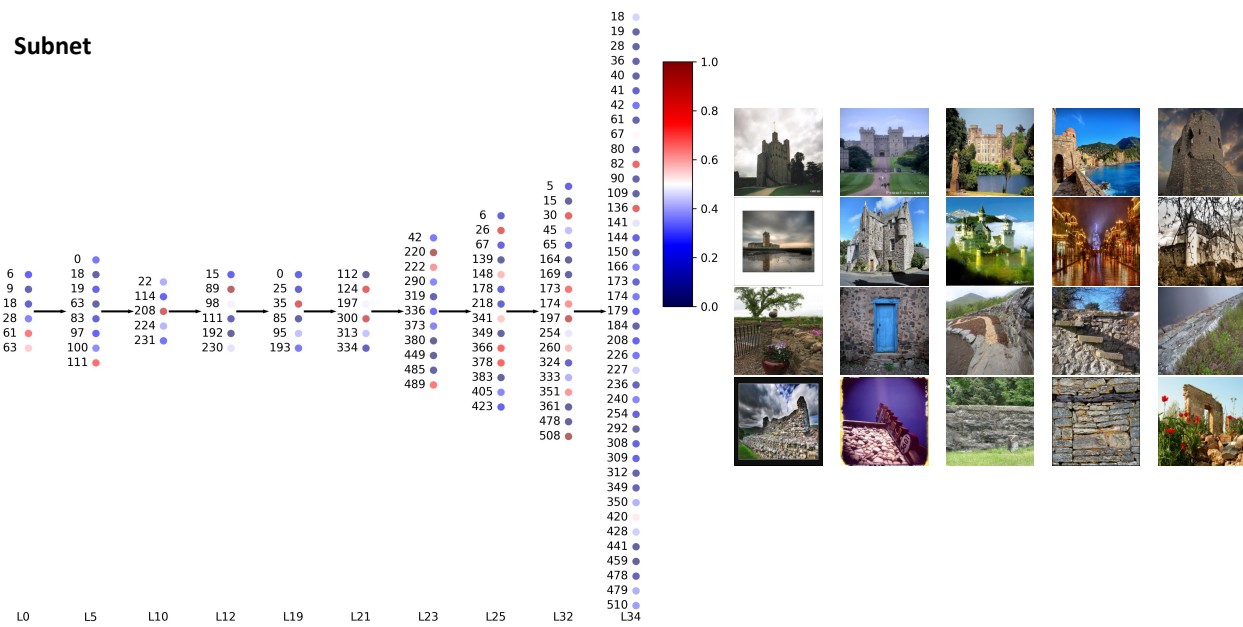

Figure 15: The extracted sub-network and 20 images from 2 classes used to extract the sub-network of case study in Figure 6.

## A.6   Case Studies

In Figure 14, the 20 images from 2 classes used to extract the sub-network of the case study in Figure 4 are displayed. All the NeuroChains are extracted from the original pre-trained model based on a few data of a subtask. The extracted sub-network and 20 images from 2 classes used to extract the sub-network of the case study in Figure 6 are shown in Figure 15.

NeuroChains can also be used to extract the sub-networks for tasks with multi-classes, and the extracted sub-network is still small enough to be analyzed and explained. In Figure 16 and Figure 17, the sub-network and the NeuroChains of a 3-classes subtask are presented. The sub-network contains 8 layers and 158 filters, which is easy to visualize. The NeuroChains corresponding to the images in the three classes are displayed.

The chains of orangutans extract features from different parts of the orangutan. The top chain of orangutan extracts the pattern in the face. L23_82 and L25_27 show the round and black node pattern. Then in L30_42 and L32_453, these nodes evolve into the pattern of eyes and noses. In L34_112, these features are integrated into the entire face of orangutan. The second chain of orangutan mainly focus on the fur and shape of arms. In L25_220, L30_439 and L32_376, the elbow part in the featuremap is highlighted, and the visualization of filters also show the pattern of curved shape of the joint part. In the first chain of tram, the pattern of the arranged windows are extracted in L25_202, L30_329 and L32_491 and the windows in the featuremap are highlighted. From the featuremap of the second chain, the bottom part of tram is highlighted and the pattern in L32_5 seems like wheels, but the visualization of other filters is not readable, which indicate that this visualization method still has limitations. In the chains of broccoli, the head part is extracted by the first chain while the stem part is captured by the second chain. From layer 10 to layer 34 of the first chain, L19_482, L23_83 and L25_247 show the pattern of dense semisphere, and L30_440, L32_281 and L34_428 look like the pattern of the head of broccoli. In the second chain, the pattern of closely arranged columns are shown in L30_421, L32_388 and L34_323, which is consistent with the shape of the stem of broccoli. From these analysis, we can find that despite the small number of filters in the sub-network, for samples from different classes, more than one critical patterns are captured to identify the samples. It shows the strong expression ability of neural network and the interpretation ability of NeuroChains.

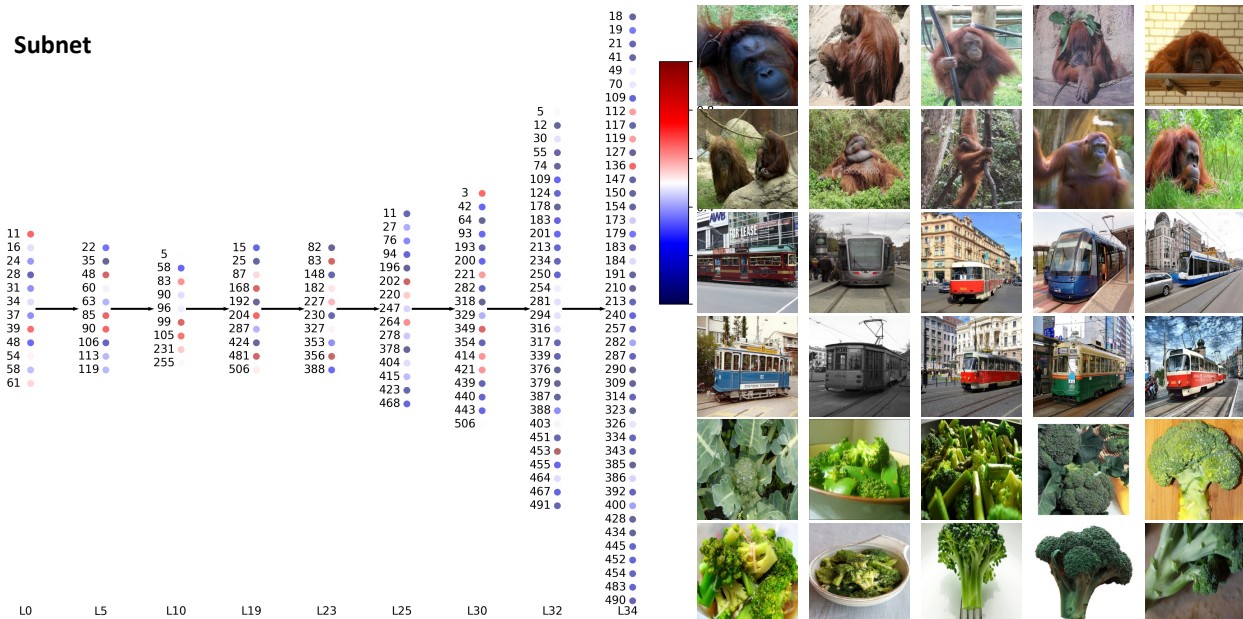

Figure 16: The extracted sub-network and 30 images from 3 classes used to extract the sub-network of case study in Figure 17.

In Figure 18, another case study to classify pineapple and leopard on Resnet-50 is shown. In the first chain, the shallow layers extract low-level patterns like edges in L3B1C2_162. L4B1C2_485, L4B1C3_384 and L4B3SC_925 from deep layers evolve these low-level patterns into the crown. In the second chain, the mid-part of the network like L3B1C1_45, L3B1C2_10, L3B1C3_541 and L4B1C1_308 show the pattern of the skin of the pineapple. Filters in the last block, i.e. L4B1C2_4, L4B1C3_1010, and L4B3SC_1106 extract the global pattern which look like the main body of pineapples. In the chains of leopard, the body and head of the leopard are lit up respectively. But the visualization of filters in the 2 chains seem similar. The skin marked with black spots is its most obvious pattern like L3B1C2_221, L4B1C1_274, L3B1C3_857 L4B1C3_462 and L4B1C3_769.

In Figure 19, a case study to distinguish indigo bunting from horse on VGG-19 is displayed. The chains of indigo bunting extract the local head pattern and global pattern of indigo bunting respectively. In the first chain, the beak and eye are lit up by L19_497, L23_375 and L25_199. L32_352 and L34_140 integrate previous pattern to the whole head. L19_44, L23_497 and L25_81 in the second chain extract the pattern of feathers. In L32_178 and L34_483, the pattern may also introduce patterns in the first chain and the global pattern including the head are lit up. Similar to indigo bunting, the chains of horse extract the head pattern and body pattern of horse respectively. L32_94 and L34_509 show the pattern like striped muscles. The whole back of the horse is highlighted. The second chain focus on the head part of the horse. L25_317 captures the mouth while L32_30 and L34_156 captures the eye. This case study shows that for some samples like indigo bunting, the global pattern is learnd by the model to identify the samples. For samples like horse, some local patterns like the body and eyes are enough.

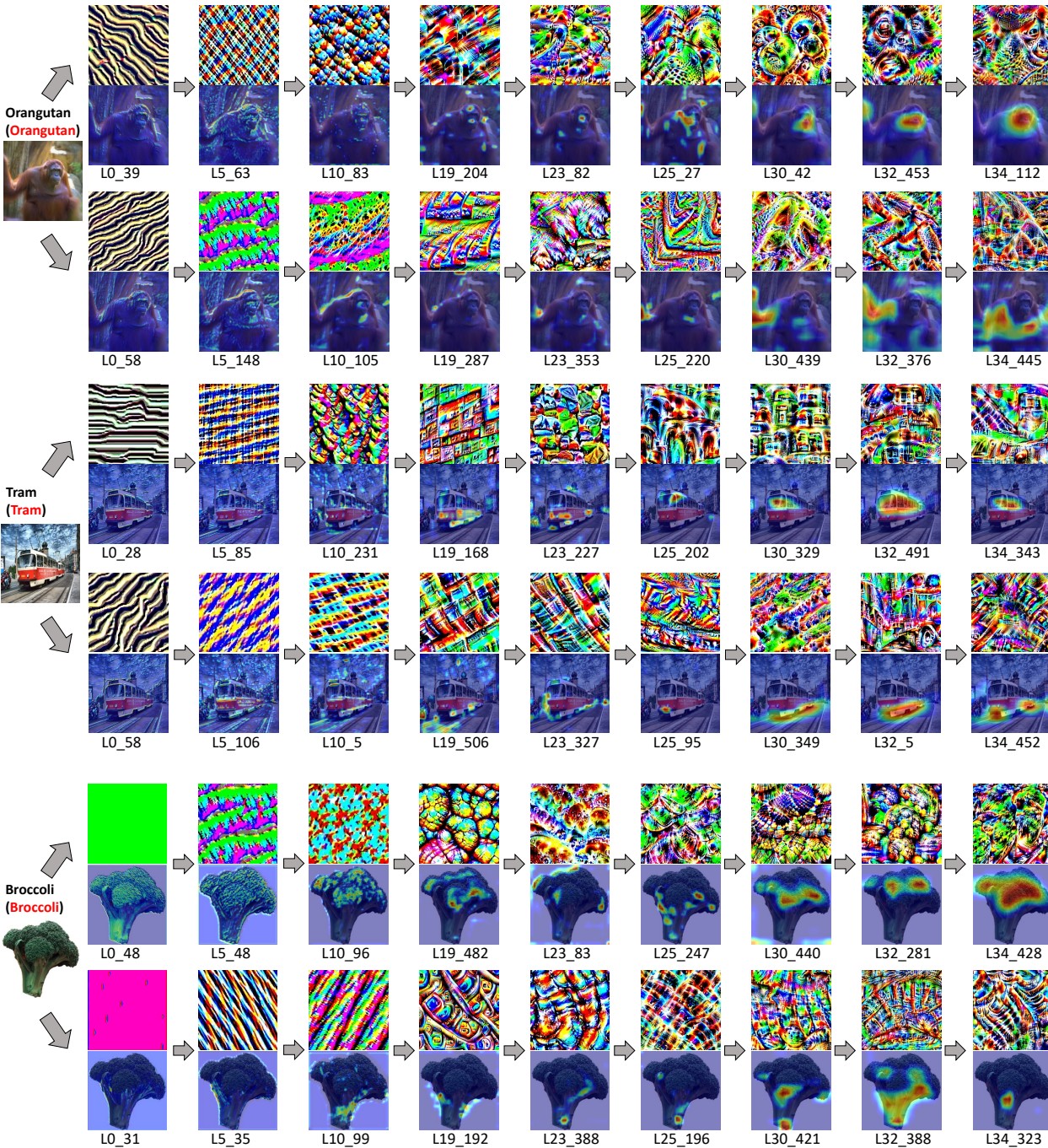

Figure 17: Inference chain by NeuroChains for VGG-19 when applied to images of 'orangutan", "tram" and "broccoli". The sub-network retains only 8/16 layers and 158/4480 filters of the VGG-19 is shown in Figure 16. Refer to Section A.6 for detailed analysis of extracted chains.

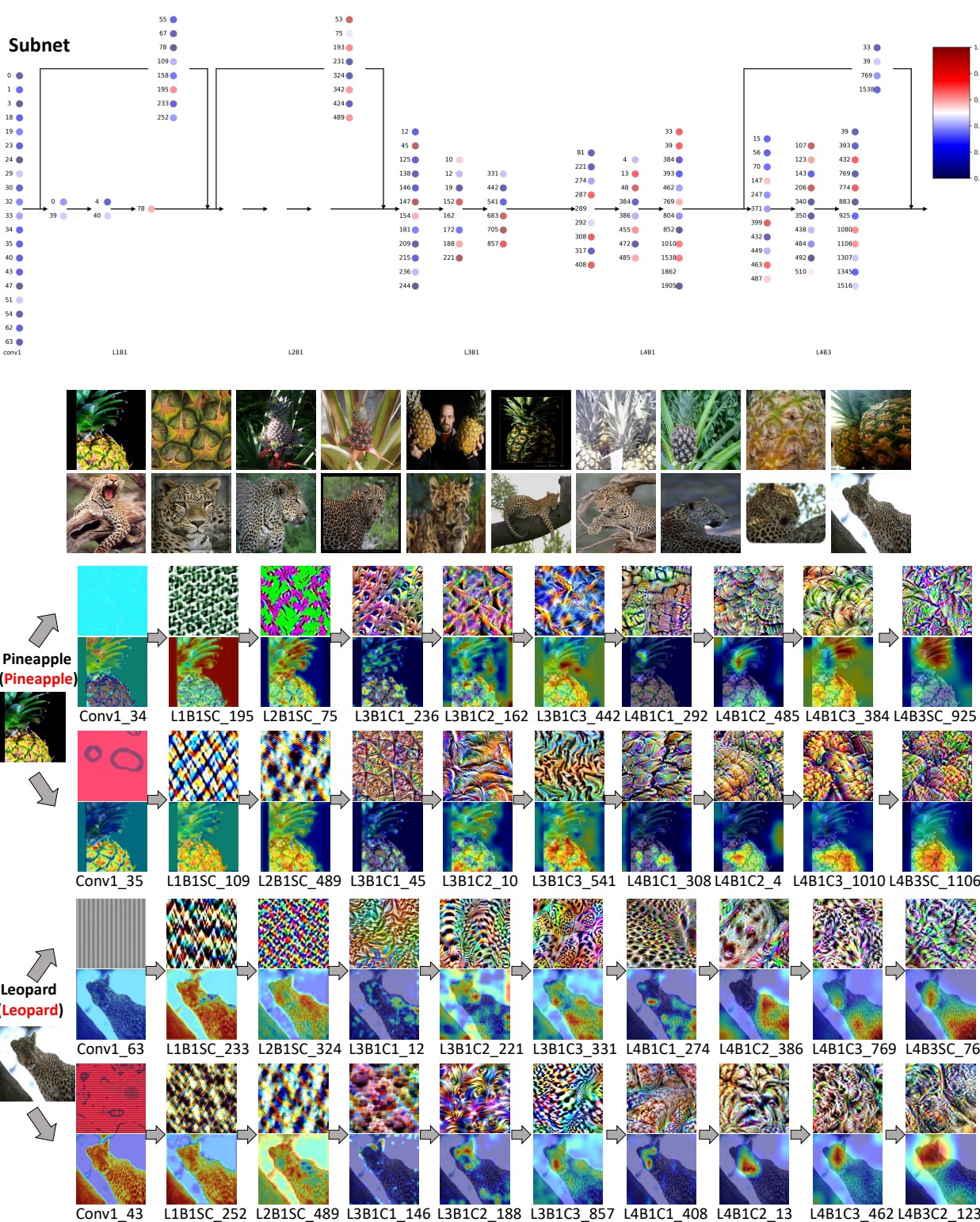

Figure 18: Inference chain by NeuroChains for ResNet-50 (pre-trained on ImageNet) when applied to 20 test images of "pineapple" and "leopard". The sub-network retains only 16/67 layers and 133/26560 filters of the ResNet-50. Refer to Section A.6 for detailed analysis of extracted chains.

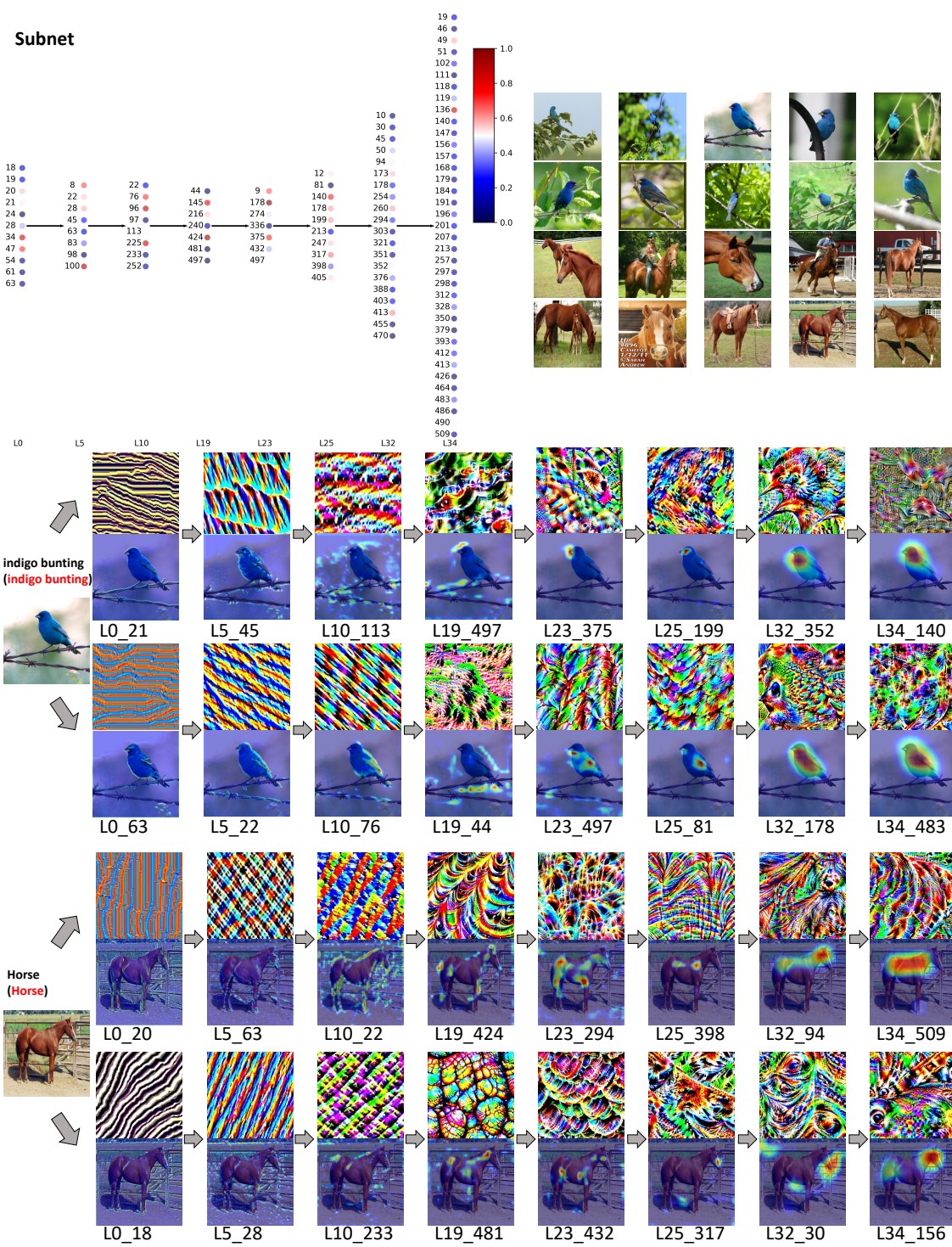

Figure 19: Inference chain by NeuroChains for VGG-19 when applied to images of "indigo bunting" and "horse". The sub-network retains only 8/16 layers and 107/4480 filters of the VGG-19. Refer to Section A.6 for detailed analysis of extracted chains.

