# OpenReview forum: "Extracting Local Reasoning Chains of Deep Neural Networks"
_TMLR — Accepted by TMLR_

### Review · Reviewer_pnfj · 2022-09-23

**Summary Of Contributions:**

The authors propose to learn masks to either remove filters or whole layers in pre-trained deep convolutional neural networks. This is done on a small subset of a two way classification problem. They find that with their method only a low percentage of filters and layers is needed to retain accuracy. Given these very small subnetwork, they visualise the features and aim to interpret the neural networks computation throughout the layer, for this "subtask".

**Broader Impact Concerns:**

Does not apply here I think.

**Requested Changes:**

1. I would propose to include "convolutional" in the title since the study, for now, is limited to this architecture type.
2. Can you include ablation studies how much your whole layer pruning adds to the sparsity / vice versa.
3. Can you show what happens to your algorithm if you prune on more classes? I think it is interesting how the NeuroChains change when doing so. Identifying chains that are important to distinguish classes from all the others for example.
4. What happens to the algorithm if you add (remove) more data? It is very surprising to me that you need so little data to train the subnetworks.
5. Can you include a detailed description of you you compute the feature scores and the heatmaps in the appendix. Or did I missed this? It is not clear to me how these are computed.

**Strengths And Weaknesses:**

I think the paper is quite well written and easy to understand. The method is simple and probably it is easy implement for a large portion of deep learning researchers and practitioners. I think it is quite surprising that only such a small subnetwork suffices to retain accuracy on a two-way-classification problem.
My problem with this is that I did not gain too much out of the visualisations / case studies proposed by the authors, on the other hand
it is possibly of limited interest how a large network distinguishes only two classes of the large dataset it was trained on although I imagine it being very difficult to do this for a larger amount of classes. See requested changes.

I am not familiar with the related work but except from insights through the case studies, which were very limited for me, I think the proposed method is "the first thing" that people would try when given a couple of training points available for pruning in contrast to unsupervised methods. Nevertheless, I think the study is quite well executed and I believe the results. Also I think the study is valuable for a larger audience and of interest to the community.

---

> ### Author Response · Authors · 2022-10-13
> **Response to Reviewer pnfj**
>
> Thanks for your comments! Please find our new experiments suggested by you and our answers to your questions below.
>
> **1. I would propose to include "convolutional" in the title since the study, for now, is limited to this architecture type.**
>
> Thanks for the suggestion! We added experimental results of NeuroChains applied to the vision transformer (ViT) in  A.1 of Appendix. It shows that NeuroChains can still produce interpretable inference chains bridging the input and its output prediction.
>
> **2. Can you include ablation studies how much your whole layer pruning adds to the sparsity / vice versa.**
>
> In fig.8 of Appendix in the revised paper, we compare the number of filters in sub-networks extracted with or without layer pruning. When layer pruning is applied, for both VGG-19 and ResNet-50, most sub-networks sizes lie in a small range $0$ to $200$. Without layer pruning, much more redundant filters will be preserved in the sub-networks. Hence, layer pruning is essential to build a small, human interpretable, and effective sub-networks.
>
> **3. Can you show what happens to your algorithm if you prune on more classes? I think it is interesting how the NeuroChains change when doing so. Identifying chains that are important to distinguish classes from all the others for example.**
>
> In Fig.17 of Appendix in the revised paper, we show a case study of applying NeuroChains to a $3$-classes classification sub-task, demonstrating the capability and effectiveness of NeuroChains on multi-classes problem. However, as classes per sub-task increase, the number of preserved filters in the sub-networks also increases, which may weaken the interpretability. We test NeuroChains on the $10$-classes classification sub-tasks. The average number of filters in the extracted sub-networks is $795(1206)$ for VGG-19(ResNet-50). Though much smaller than the original network, the extracted sub-networks are too large to explain.
>
> That being said, NeuroChains can explain the correlation among multiple classes by identifying the chains important to distinguish a class from the others. We conduct a binary-classification experiment shown in Fig.10 with a detailed analysis in A.3 of Appendix in the revised paper, please refer to it.
>
> **4. What happens to the algorithm if you add (remove) more data?**
>
> Fig.9 in Appendix of the revised paper shows how the sub-networks' validation-set accuracy changes when increasing the training samples per class in each sub-task. For both pre-trained VGG-19 and ResNet-50, when the number of samples per class is too small ($\leq5$), the sub-network tends to be overfitting and cannot generalize well to unseen validation data. However, when the number of samples per class increases to $10$, the sub-network starts to achieve promising validation-set accuracy. If further increasing the samples per class, the validation-set accuracy quickly saturates. Therefore, in the experiments, $10$ samples per class turn to be the sweet spot to extract sub-networks with sufficiently good generalization performance so they can be interpreted as the inference chains of the pre-trained model on given sub-tasks.
>
> **5. Can you include a detailed description of you compute the feature scores and the heatmaps in the appendix.**
>
> To compute the scores, we firstly initialize each filter in the pre-trained model using Eq.3 by multiplying it with a learnable score. During pruning, only the scores are optimized (filters are frozen). At the end of pruning, only filters with large scores are preserved. The feature scores you mentioned should be the optimized filter scores at the end of pruning, which aims to measure the importance of each filter for the sub-task.
>
> To obtain the heatmaps, each filter applied in the sub-network produces a featuremap for an input image. By applying an activation function to the featuremap, only a few pixels are activated, and they highlight the regions in the featuremap, which represent what features/patterns of the input sample are detected by the filter. Since the featuremap size is always smaller than the input image, we apply the bilinear interpolation to upsample the featuremap up to the input image size. Finally, we overlap the resized featuremap with the input image, which results in a heatmap highlighting the regions of the input image represented by the corresponding filter. This part is added to the Appendix of the revised paper.

---

### Review · Reviewer_JDVr · 2022-09-23

**Summary Of Contributions:**

NeuroChains optimizes differentiable scores on filter outputs and neural network layers, to select a sub-network of a neural network that has predictions matching that of the neural network. There is no fine-tuning of the weights of the neural net.

The hypothesis of the papers is that while neural networks can be large, when applied to a sample, the "inference" relies only a small subset of layers and filters.

The authors confirm that hypothesis by evaluating trained VGG-19 and ResNet-50 neural networks. The authors show the importance of filters, for example removing only one highly scored filter from the sub-network can significantly degrade performance. Then the authors confirm with analysis that neural networks are making gradually more and more concrete features for a particular class.

**Broader Impact Concerns:**

None.

**Requested Changes:**


1. In general, I think it would be interesting to see more discussion about what are meaningful tasks to be studied with NeuroChains.

2. Right now your method is focused on categorical datasets. Could you elaborate on potentially other types of datasets?

3. Please make a comparison between the inferences of different types of neural networks (in the way they are designed or trained). I think the neural networks should not be studies in isolation, when NeuroChains is applied.

4. How does your method relate to the Lottery Ticket Hypothesis, which asks you to fine-tune the weights from the original initialization? Do you think you are finding new types of Lottery Tickets here? And if so, how are they different from the tickets we are familiar with? Could you elaborate?

Minor:

* in abstract: extra space before a period

* in first paragraph of Intro: despite its <- despite theirs

* 2nd paragraph of Intro: an given <- a given

* whether removing the entire layer <- whether to remove the entire layer

There are more grammatical errors and typos throughout the text. Please, proofread the text carefully.

**Strengths And Weaknesses:**

Strengths:

1. NeuroChain is a well-executed and documented methodology for finding sparse subnetworks that can elucidate the inference which a neural network performs during a forward pass.

2. The approach could potentially be applied to any neural network. Thus, it is a general method in spirit.

3. While simple, the method is distinguished from the existing literature.

Weaknesses:

1. The paper only studies ResNet-50 and VGG-19. I would argue that for a modern analysis of neural networks, it would also be interesting to study transformers, e.g. ViT. Trained ViTs are publicly accessible, so I think a study on ViTs (and comparison with ResNet-50 and VGG-19) should be feasible.

2. The Introduction should be more specific what is meant by (sub)tasks.

3. Right now the study is focused on particular neural networks in isolation. It would have been more interesting to consider comparison between different architectures using your tool.

---

> ### Author Response · Authors · 2022-10-13
> **Response to Reviewer JDVr**
>
> Thanks for your comments and your efforts for reviewing our paper! In the following, we provide answers to your questions.
>
> **1. It would be interesting to see more discussion about what are meaningful tasks to be studied with NeuroChains. What is meant by (sub)tasks**
>
> A task can be decomposed into simple sub-tasks, e.g., a sub-task of a classification task can be defined as classification on a small subset of classes. NeuroChains mainly focuses on these simple sub-tasks and interpreting how they are performed in a pre-trained network. We keep each sub-task simple (a small subset of classes) so the sub-network and its reasoning chains are sufficiently small to be visualized and understood by humans.
>
> For example, in this paper, the original networks are pre-trained on ImageNet for a 1000-classes classification task, while each sub-task is a binary or three-classes classification task sampled from the 1000 classes.
>
> **2. Could you elaborate on potentially other types of datasets?**
>
> Our method can be applied to other types of datasets, though the experiments mainly focus on classification tasks.
>
> For a non-classification task, we only need to replace the loss in Eq.(5) with the task's loss in order to measure the mismatch between the sub-network and the pre-trained network. We will investigate other types of tasks in our future work.
>
> **3. Make a comparison between the inferences of different types of neural networks.**
>
> To compare different types of neural networks, we additionally apply NeuroChains to the vision transformer (ViT). The case study and detailed analysis are shown in A.1 of Appendix in the revised paper.
>
> From the case study, we can find that the inference process of ViT and CNN are very different.
> In CNN, from the shallow layers to the deep layers, the model first extracts the low-level local patterns in the input sample and then aggregates them to get high-level global patterns step by step to classify the input sample.
> In ViT, the model can capture both the local and global patterns of the input since the very bottom blocks/layers. During inference, the model gradually identifies patches important to the output and mainly extracts patterns relevant to the class token.
>
> **4. How does your method relate to the Lottery Ticket Hypothesis.**
>
> Our method differs from the Lottery Ticket Hypothesis (LTH) in two folds.
>
> (1) The objectives.
> LTH aims to find a sparse sub-network architecture that can be trained from initialization to reach comparable performance as the pre-trained network.
> In contrast, each sub-network extracted by NeuroChains has to keep the original filters of the pre-trained network but only optimize their weights. Hence, NeuroChains can be used to explain the pre-trained model's inference process on the given sub-task.
>
> (2) The initialization of the sparse network.
> In LTH, the winning ticket starts training from scratch.
> In NeuroChains, each sub-network consists of pre-trained filters copied from the pre-trained model, which accelerate the finetuning.

---

### Review · Reviewer_nEr7 · 2022-09-28

**Summary Of Contributions:**

This paper presents a new setting for explainable AI. Concretely, the authors propose to explain the inference process of pretrained deep learning via network pruning. Concretely, pruning the network down to a tiny subnetwork, whose intermediate feature maps can be used to study and explain the reasoning process of the original network. Different from compression, this paper 1). does not allow fine-tuning and only match the output distribution between original and sub networks on subtasks, defined as a subset of labels in this work. The specific pruning method is to optimizer a learnable mask on the model weight, and perform pruning based on the resulting scores.

Leveraging the proposed tool, the author provides case studies on ResNet and VGG to visualize their inference process.

**Broader Impact Concerns:**

Since the subnetworks will be used to analyze and interpret the original model, perhaps the author could discuss any potential biases that might be introduced in the process of generating these subnetworks.

**Requested Changes:**

N/A

**Strengths And Weaknesses:**

Strength:

- The paper presents an interesting alternative setting (and method) to explainable AI. Specifically it studies explaining network prediction on a subtask, rather than on a single example as prior works.
- Detailed analysis on common architectures are presented.


Weakness / Concerns:
- My main conern is that, since the pruning objective is defined on a subtask, is it possible that the pruning process "overfits" to the subtask? If so, this might affect the reliability of the resulting subnetworks in representing the model's actual reasoning chain. To what extend can we measure and trust the representability of these subnetworks? Although the weights of these subnetworks directly come from the original model, the pruning process along might still induce biases.

---

> ### Author Response · Authors · 2022-10-13
> **Response to Reviewer nEr7**
>
> Thanks for your comments and your efforts for reviewing our paper! In the following, we provide answers to your questions.
>
> **1. Is it possible that the pruning process "overfits" to the subtask? ... To what extend can we measure and trust the representability of these sub-networks?**
>
> We keep the pruning from overfitting by preserving exactly the same filters of the original network in the sub-networks without changing their parameters. The only learnable parameters in the pruning process are the scores for filters, which are much less than the frozen parameters, for example, their ratio is 5516:143667240(37836:25557032) for VGG-19(ResNet-50). By limiting the degree of freedom in pruning, our method dose not suffer from overfitting.
>
> There are two methods we use to measure the representability and trustworthiness of the sub-networks:
>
> (1) We evaluate the sub-networks' generalization performance on the validation sets of sub-tasks. For example, the validation accuracy is $85.10\pm7.50$%($83.33\pm6.33$%) for VGG-19(ResNet-50), which indicates that the sub-networks do not overfit to their training data but generalizes promisingly to unseen validation data.
>
> (2) We evaluate the Pearson correlation between the output probabilities of the original pre-trained model and the sub-networks on the validation sets of sub-tasks. Their correlation on VGG-19 and ResNet-50 are $0.83$ and $0.80$ respectively (with nearly zero p-value), indicating a strong correlation. Hence, the sub-networks are not overfitting in the pruning process but faithfully preserve the output distributions of the original network.

---

### Decision · Action_Editors · 2022-11-12

**Recommendation:** Accept as is

**Comment:**

The paper proposes a framework for interpreting deep neural networks focused on extracting a very small (e.g. 150 filters out of 4480 filters of VGG-19) sub-network critical dedicated to some aspect of the training task, such as distinguishing between two selected classes from the dataset.

After discussions, the reviewers were unanimously supportive of accepting the paper. They highlighted several strong aspects of the submission:

1. It has the potential to become a new useful tool in the interpretability toolbox. Importantly, experiments are thorough enough to warrant optimism about this direction. As written by Reviewer nEr7 “I think this paper presents a novel aspect to the XAI community. The experiments and discussions are sufficient to justify the objective of this paper.”. Reviewer JDVr also appreciated that the framework seems to be novel.
2. As highlighted by reviewer pnfj, the result that only a very small network is sufficient for distinguishing between two classes is quite interesting on its own.
3. The method can be in principle applied to any architecture. In particular, the Authors have added the revision experiments on ViT. The experiments suggest that ViT and CNN arrive at the final decision using a significant difference reasoning mechanism, which again is an interesting result on its own (even if likely known in the field).

Weaker sides of the submission include the lack of head to head comparison to other interpretability methods. While it is challenging to compare interpretability methods, there are certain experiments one could do. For example, one could use the interpretability method in some downstream task (e.g. identifying issues in a dataset). The experiments are mostly narrowed down to CNNs (exception is the ViT experiments in the Appendix).

All in all, it is my pleasure to recommend acceptance of the paper. Thank you for submitting your work to TMLR. Please make sure you address all comments by the reviewers. I would also encourage, though it is not of course required, to promote ViT experiments to the main text, as these experiments highlight the strong sides of the framework.

**Audience:**

The paper could be interesting to a wide audience, especially those interest in the interpretability of deep neural networks.

**Claims And Evidence:**

The paper is supported by sufficient evidence for all major claims made in the paper.